# WHICH ENGLISH DO LLMs PREFER? TRIANGULATING STRUCTURAL BIAS TOWARD AMERICAN ENGLISH IN FOUNDATION MODELS

## ABSTRACT

Large language models (LLMs) are increasingly deployed in high-stakes domains, yet they expose only limited language settings, most notably "English (US)", despite the colonial history and global diversity of English. We interpret dialectal asymmetries through a holistic postcolonial lens, showing they emerge not only as downstream failures but as structural artifacts of the LLM development pipeline itself. Using a curated lexicon of 1,813 American–British variants, we triangulate evidence across three stages: (i) audits of six major pretraining corpora reveal systematic skew toward American English, (ii) tokenizer analyses demonstrate that British forms incur higher segmentation costs, and (iii) generative evaluations with our proposed DiAlign metric show consistent preference for American variants. This constitutes the first systematic and multi-faceted examination of dialectal asymmetries in standard English varieties across the phases of LLM development. We find that these models exhibit structural bias that privileges American English as the de facto norm, shaped by geopolitical histories of data curation and linguistic standardization. Our study raises concerns of linguistic homogenization, epistemic injustice, and inequity in global AI deployment, while offering practical guidance for developing more dialectally inclusive language technologies.

## 1 INTRODUCTION

The United Nations affirms *language rights* as fundamental, with Article 19 of the Universal Declaration guaranteeing the freedom to communicate in one's language of choice. Modern software systems operationalize this principle through localization (Southwell, 2021), offering explicit language settings. Large language models (LLMs), increasingly deployed as software-as-a-service in domains such as education (Shahzad et al., 2025), law (Lai et al., 2024), and public administration (Kulal et al., 2024; Madan & Ashok, 2023), often lack such flexibility. Despite evidence that English is the most effective prompting language (Behzad et al., 2024), widely used platforms such as ChatGPT and Claude expose only "English (US)" as a selectable option. As governments and institutions adopt these models for administrative processes and public service delivery, the privileging of a single variety of English acquires systemic significance. This raises foundational questions: Which forms of English do LLMs prefer, what are the implications for fairness, efficiency, and inclusion, and can (or should) these preferences be redirected? We address these questions by systematically examining two dominant standard varieties, American English (AmE) and British English (BrE), through a holistic postcolonial lens that positions AmE as the digitally dominant and structurally advantaged default and BrE as a widely institutionalized yet comparatively marginalized variety, and by analyzing how linguistic asymmetries are encoded and propagated across the entire LLM development pipeline.

English is the most widely used international language, serving as an official or special language in over 75 countries and spoken by more than 1.5 billion people worldwide (Galan, 2025). Its global dominance reflects *two trajectories*: British colonial expansion, which entrenched English in governance and education across Africa, Asia, and other regions, and twentieth-century American hegemony, which spread English through commerce, media, and technology (Crystal, 2003; Nordquist, 2024). This history produced diverse English varieties shaped by local identities (Trudgill, 2000), yet sociolinguistic research shows that power and prestige determine which forms are legitimized or marginalized (Labov, 1972; 2006). Divergences between AmE and BrE span spelling,

vocabulary (Table 1), grammar, structure, idioms, style, and pronunciation (Liz, 2024).[1] BrE retains normative prestige in many former colonies (Figure 1), including South Asia, Nigeria, and Singapore, where it remains embedded in governance, education, and law.[2] It is also the standard of EU institutions and underpins "Commonwealth English" (Calabrese et al., 2015), actively promoted by the UK across more than 100 countries[2] through initiatives such as the Oxford Dictionary, the British Council, and IELTS. AmE, by contrast, dominates global culture and digital communication through Hollywood, music, mass media, and technological platforms (Gonçalves et al., 2018), positioning it as a de facto global norm. The authority of these standards derives not from linguistic merit but from sociopolitical power (Milroy & Milroy, 1999; Lippi-Green, 2012), creating a dynamic where two influential dialects coexist, each exerting distinct cultural and normative influence. While other Englishes exist[3], this paper focuses on AmE and BrE as the two dominant postcolonial standards.

Central to the success of LLMs is their training on massive corpora drawn largely from the internet, where English dominates, accounting for roughly 50–60% of global web content (Dodge et al., 2021; Petrosyan, 2025). Although dataset compositions are often undisclosed, available evidence indicates that English constitutes about 92.65% of GPT-3's training data,[4] 89.7% of Llama 2's (Touvron et al., 2023), and nearly 90% of Claude 2's (Anthropic, 2023). For AmE and BrE specifically, the abundance of digitized resources rules out scarcity as a limiting factor. The critical question, then, is which form of English these models preferentially learn, encode, and propagate. While reliance on English-heavy corpora reflects its global dominance, it also foregrounds an underexplored dimension: whether LLMs reproduce asymmetries between AmE and BrE rooted in distinct historical and sociopolitical trajectories. This paper investigates how such dynamics manifest across the LLM development pipeline, examining whether and how models exhibit preferences between AmE and BrE and what those preferences reveal about broader socio-technical biases. This study is driven by an intriguing question: *Which English variety do LLMs implicitly privilege, and with what consequences?*

We seek to identify a root cause: the presence of *structural bias*, a specific and under-studied form of linguistic bias and, to our knowledge, the first systematic and multi-faceted examination of its impact on standard English varieties, wherein language technologies, by design, may favor certain languages, dialects, or sociolects over others (Bender et al., 2021). Such biases can lead to *epistemic injustice* (Fricker, 2007), where marginalized linguistic communities are systematically underrepresented in algorithmic systems (Helm et al., 2024). This distinction is critical: if LLMs implicitly treat AmE as the default or normative form, it raises profound concerns for equitable AI, potentially resulting in linguistic homogenization and degraded user experiences for speakers aligned with British English norms. By interpreting our analysis through a postcolonial lens (§3), we highlight how geopolitical histories of data curation, digital dominance, and linguistic standardization shape pretraining corpora, tokenizers, and generative behaviors of modern LLMs. Rather than documenting performance disparities solely as downstream failures (Ziems et al., 2022; Fleisig et al., 2024), our study probes their root causes by triangulating across the entire LLM development pipeline (data → tokenization → generation). Concretely, our investigation centers on three core research questions:

---

**Research Questions**

**RQ1:** To what extent do large-scale pretraining corpora skew toward American over British English? We provide corpus-level audits of major LLM pretraining datasets to quantify dialectal imbalance in token distributions (Section 5).

**RQ2:** How do regional tokenizers encode AmE and BrE variants, and what does this reveal about dialectal representation? We examine subword-level disparities across tokenizers developed in American, European, Chinese, and postcolonial contexts (Section 6).

**RQ3:** Do LLMs exhibit generative preferences for AmE over BrE? We assess output dialectal preferences under contextual prompts, using the proposed DIALIGN score to estimate alignment across lexical, grammatical, structural, stylistic, and multi-word contrasts (Section 7).

---

[1] wiki/Comparison_of_American_and_British_English, see Table 8 and Table 9 for illustrative examples.

[2] en.wikipedia.org/wiki/British_English, see Appendix A for a brief historical background.

[3] Canadian English blends BrE and AmE influences due to its history and geographical proximity but also has unique features; Indian, Australian, and New Zealand English largely inherit BrE (Acolad, 2020; Liao, 2023).

[4] OpenAI GPT-3 Dataset Language Statistics (GitHub, accessed April 26, 2025)

## 2 RELATED WORK

**Pretraining data audits and curation.** Nearly all advanced model capabilities originate from the scope and composition of pretraining data, motivating a growing body of work on auditing and curation. Analyses highlight how dataset age, coverage, and quality affect generalization (Longpre et al., 2024), while audits reveal duplication, contamination, and provenance gaps in widely used corpora (Elazar et al., 2024; Longpre et al., 2025). Beyond audits, strategies for improving data utility include practical construction recipes for large-scale corpora (Parmar et al., 2024), register- and domain-aware sampling (Myntti et al., 2025), and recycling filtered web text (Nguyen et al., 2025). Our study extends this line of work by foregrounding American vs. British English as a dimension of representational skew, showing how such imbalances can cascade into tokenization disparities and ultimately shape the generative behavior of LLMs.

**Tokenizer fairness.** Biases can arise *before* generation, at the subword segmentation stage. Prior work shows that semantically equivalent strings can receive uneven tokenization across languages, with consequences for efficiency, context budget, and cost (Petrov et al., 2023). Recent work quantifies the *causal* impact of uneven tokenization, showing that collapsing a multi-token span into a single token can inflate a word's probability by more than an order of magnitude (Lesci et al., 2025). Complementary work proposes Parity-Aware Byte-Pair Encoding, which slightly relaxes compression to equalize token counts across languages and improve cross-lingual fairness (Foroutan et al., 2025). Tokenization length further correlates with demographic attributes of personal names, reinforcing or even creating social biases (An & Rudinger, 2023), while small lexical alternations, such as brand vs. generic drug names, expose fragility in LLM representations (Gallifant et al., 2024). In machine translation, subword design and training distribution jointly amplify gender bias, with female and non-stereotypical forms more often fragmented (Iluz et al., 2023). We extend this line of inquiry to *intra-English* dialects, showing that tokenizers encode uneven segmentation for dialectal variants.

**Dialect robustness in NLP tasks.** Work on fairness has shown that dialectal variation, especially in African American English (AAE) and South Asian Englishes (SAsE), can yield systematic performance gaps across core NLP tasks such as tagging, classification, and sentiment analysis (Jørgensen et al., 2016; Blodgett et al., 2016; Kiritchenko & Mohammad, 2018). Recent studies further reveal that LLMs encode negative stereotypes toward AAE (Hofmann et al., 2024; Fleisig et al., 2024) and that SAsE speakers often perceive NLP systems as brittle or exclusionary (Holt et al., 2024). Frameworks such as Multi-VALUE highlight robustness gaps across dialects (Ziems et al., 2023), but even standard varieties like AmE and BrE remain underexplored. Our work addresses this gap by probing the root causes of AmE–BrE variation across the entire LLM development pipeline, interpreting it through a postcolonial lens as, to our knowledge, one of the earliest systematic examinations of dialectal asymmetries. An extended discussion of related work is provided in Appendix K.

## 3 INTERPRETING STRUCTURAL BIAS THROUGH A POSTCOLONIAL LENS

Postcolonial theory studies how power relations created by colonialism persist after formal empire, shaping language, culture, and knowledge in both formerly colonized states and former imperial centers (Bhabha, 1994; Schneider, 2007). The global spread of English was inseparable from British colonial expansion: BrE was installed as the language of administration, education, and law across large parts of Africa, Asia, the Caribbean, and the Pacific, and often persisted as the official or de facto standard after independence (Figure 1). It continues to hold normative prestige in many former colonies, across much of the Commonwealth, and in key European Union institutions.[2]

By contrast, AmE dominates mass media and digital communication, and LLMs trained on web-scale internet data are likely to inherit its norms. We ask which variety of English LLMs preferentially learn, encode, and propagate, focusing on two dominant postcolonial standards, AmE and BrE, whose institutional status enables a controlled, high-precision comparison. Systematic privileging of AmE has downstream implications also for other postcolonial Englishes that build on BrE, such as Indian, Nigerian, and Australian English (Acolad, 2020; Liao, 2023), and raises inclusivity concerns when users expect BrE-aligned norms, especially in education, journalism, government, and legal texts.

Our holistic postcolonial perspective interprets this dialectal skew as a manifestation of *structural bias*, a systematic preference for particular standard varieties, and employs this framing to analyze

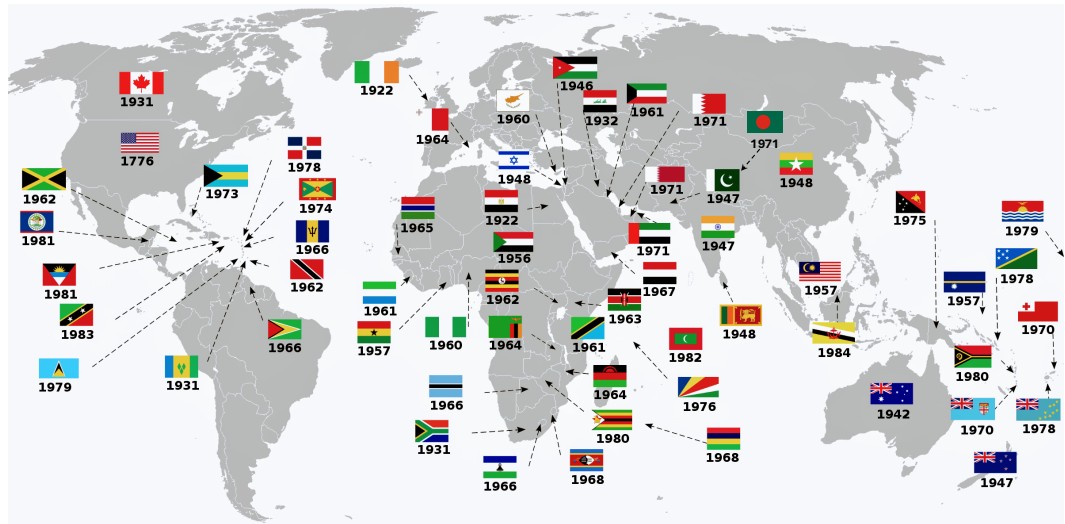

Figure 1: Timeline of independence across countries formerly under British colonization. The map highlights the wave of decolonization in the mid-twentieth century, when nations in Africa, Asia, the Caribbean, and the Pacific gained sovereignty. This geopolitical shift marked the decline of direct colonial governance but reinforced the institutional legacy of *British English (BrE)* in education, government, journalism, and law across many of these regions (Tikly, 2016; Phillipson, 2018).

the consequences of such dialectal asymmetries for global inclusivity. We triangulate evidence across the entire LLM development pipeline to surface these structural biases: ① pretraining corpora (§5), ② tokenizer representations (§6), and ③ generative preferences (§7). This triangulation allows us to trace how dialectal asymmetries are introduced, amplified, and manifested in outputs, linking empirical audits of LLM behavior to broader concerns about linguistic homogenization and epistemic injustice in global AI deployment, and motivating component-wise design recommendations for dialect-sensitive corpus construction and filtering, tokenizer design, alignment, and evaluation (§8).

## 4 DIALECTAL VARIANT CORPUS: TYPOLOGY OF AME–BRE LEXICONS

To operationalize our study of dialectal preferences in LLMs, we construct a curated corpus of 1,813 parallel lexical variants between AmE and BrE. This resource is designed to serve as a reference set of dialectal markers for consistent analysis across the research questions (Sections 5 to 7).

The variant pairs were manually compiled from authentic linguistic sources and web-based lexicons (Table 7). We merged data from multiple sources and removed duplicates to form a unified lexicon. To ensure linguistic comparability and analytical precision, we retained only strict one-to-one word-level mappings, and excluded many-to-one (e.g., *"drug store"* (AmE) vs. *"chemist's"* (BrE)), one-to-many (e.g., *"restroom"* (AmE) vs. *"public toilet"* (BrE)), and many-to-many cases (e.g., *"parking lot"* (AmE) vs. *"car park"* (BrE)). This constraint aligns with our goal of treating words as atomic units, since words, when tokenized, form the basic building blocks of LLMs. Restricting to one-to-one mappings ensures consistency across analyses and is essential for the tokenizer study [RQ2 (§6)], where precise word-level comparisons are required to directly compare segmentation behavior.

The resulting lexicon spans both orthographic (spelling-based) and lexical (vocabulary-based) differences. Table 1 presents an overview of the typology and distribution of variation types in the corpus, with representative examples. Details on the categorization schema and on the data sources used to construct the variant corpus are provided in Appendix B and Table 7, respectively.

## 5 RQ1: AUDITING DIALECTAL SKEW IN PRETRAINING CORPORA

To empirically ground our investigation of dialectal structural bias in LLMs, we begin by auditing six major open-access pretraining corpora for statistically significant skew in AmE versus BrE usage.

Table 1: Distribution of preferred $1{,}813$ AmE (🇺🇸) and BrE (🇬🇧) variant pairs across common linguistic categories from the curated corpus. We report the percentage of total entries and representative examples per category, grouped into orthographic (*spelling*) and vocabulary-based differences.

| Category | Difference Type | % of Pairs | Examples | |
|---|---|---|---|---|
| | ends in "-or" (AmE) **vs.** "-our" (BrE) | 2.26% | color (🇺🇸) **vs.** colour (🇬🇧) | labor (🇺🇸) **vs.** labour (🇬🇧) |
| | ends in "-ize" (AmE) **vs.** "-ise" (BrE) | 11.58% | organize (🇺🇸) **vs.** organise (🇬🇧) | realize (🇺🇸) **vs.** realise (🇬🇧) |
| | ends in "-er" (AmE) **vs.** "-re" (BrE) | 1.65% | center (🇺🇸) **vs.** centre (🇬🇧) | liter (🇺🇸) **vs.** litre (🇬🇧) |
| **Orthographic/** | ends in "-og" (AmE) **vs.** "-ogue" (BrE) | 0.55% | dialog (🇺🇸) **vs.** dialogue (🇬🇧) | catalog (🇺🇸) **vs.** catalogue (🇬🇧) |
| **Spelling** | ends in "-ense" (AmE) **vs.** "-ence" (BrE) | 0.22% | defense (🇺🇸) **vs.** defence (🇬🇧) | pretense (🇺🇸) **vs.** pretence (🇬🇧) |
| | "e" (AmE) **vs.** "ae" (BrE) | 4.03% | esthetic (🇺🇸) **vs.** aesthetic (🇬🇧) | pediatric (🇺🇸) **vs.** paediatric (🇬🇧) |
| | words with single "l" **vs.** double "l" | 8.88% | traveler (🇺🇸) **vs.** traveller (🇬🇧) | enroll (🇺🇸) **vs.** enrol (🇬🇧) |
| | sublexical spelling variation | 49.75% | jewelry (🇺🇸) **vs.** jewellery (🇬🇧) | program (🇺🇸) **vs.** programme (🇬🇧) |
| **Vocabulary** | different lexical items entirely | 21.07% | elevator (🇺🇸) **vs.** lift (🇬🇧) | flashlight (🇺🇸) **vs.** torch (🇬🇧) |

Table 2: AmE vs. BrE variant usage across six pretraining corpora, segmented into orthographic (spelling) and vocabulary-based differences. Each entry reflects the probability of observing either the AmE or BrE variant for a given word pair. We aggregate these statistics across all 1,813 pairs to yield corpus-level dialectal distributions, defining a probability distribution over mutually exclusive outcomes. All results are statistically significant under the Wilcoxon Signed-Rank Test ($p$-value $< 0.01$). Datasets marked with * denote sampled subsets. RedPajama and Dolma include mixed-domain content (e.g., </> code, 📄 papers, 📑 forums, 💬 social media). All probabilities are shown as percentages. LLaMA tokenizer (Grattafiori et al., 2024) was used to compute token statistics.

| Data Source | Document Type | Documents (*millions*) | Tokens (*billions*) | Orthographic | | Vocabulary | |
|---|---|---|---|---|---|---|---|
| | | | | AmE (🇺🇸) | BrE (🇬🇧) | AmE (🇺🇸) | BrE (🇬🇧) |
| Book Corpus (2015) | 📗 books | 74 | 1.28 | 86.81 | 13.19 | 75.00 | 25.00 |
| Wikipedia (2024) | 📘 encyclopedic | 6.4 | 4.3 | 72.94 | 27.06 | 61.43 | 38.57 |
| Common Crawl (C4) (2020) | 🌐 web pages | 365 | 156 | 75.12 | 24.88 | 67.00 | 33.00 |
| Falcon RefinedWeb (2023) | 🌐 web pages | 968 | 600 | 77.34 | 22.66 | 68.35 | 31.65 |
| RedPajama* (2024) | 📗📘🌐</>📄📑 mixed | 0.93 | 1.0 | 76.03 | 23.97 | 66.05 | 33.95 |
| Dolma* (2024) | 📗📘🌐</>📄💬 mixed | 14.28 | 10 | 77.30 | 22.70 | 67.77 | 32.23 |

Leveraging our curated set of 1,813 AmE–BrE lexical variant pairs (§4), we compute variant-specific token distributions to quantify the extent and direction of dialectal imbalance (see Appendices G and I for details). These lexical markers not only capture surface-level contrasts but also provide reliable signals of surrounding structural and stylistic tendencies. We refer to any such consistent asymmetry as *dialectal skew*, which we interpret as indicative of *structural bias* in pretraining corpora.

**Methodology**  For each corpus, we extract raw frequencies $f_{\text{AmE}}$ and $f_{\text{BrE}}$ corresponding to each word pair. To normalize and quantify dialectal usage, we compute a probability distribution:

$$P_{\text{AmE}} = \frac{f_{\text{AmE}}}{f_{\text{AmE}} + f_{\text{BrE}}}, \quad P_{\text{BrE}} = \frac{f_{\text{BrE}}}{f_{\text{AmE}} + f_{\text{BrE}}}.$$

These probabilities represent the likelihood of observing either variant within a pair and define a valid distribution over mutually exclusive outcomes. We aggregate these statistics across all word pairs to yield corpus-level dialectal distributions, stratified by orthographic and vocabulary-based categories. To assess the statistical significance of observed directional bias, we apply the Wilcoxon Signed-Rank Test to the pairwise frequency differences ($f_{\text{AmE}} - f_{\text{BrE}}$). This non-parametric test is well suited for skewed and zero-inflated distributions typical of large-scale language corpora (Dror et al., 2018). All corpora yielded $p$-values below 0.01, confirming significant deviation from dialectal parity.

**Results & Analysis**  Table 2 reports corpus-level dialectal distributions. All six datasets exhibit a statistically significant skew toward AmE, particularly in orthographic variants (e.g., *color* vs. *colour*), where AmE spellings dominate with margins exceeding 70%. Vocabulary-based differences (e.g., *elevator* vs. *lift*) show a less extreme, but still consistent, AmE preference. These findings demonstrate that dialectal skew is not incidental but structurally embedded in the pretraining datasets that serve as the backbone of modern LLMs.

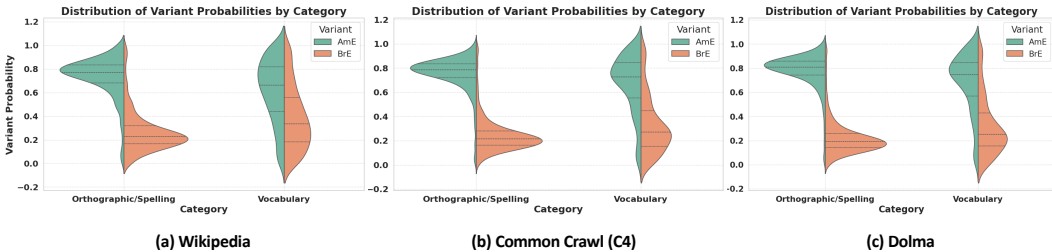

Figure 2: Violin plots showing the distribution of AmE vs. BrE variant probabilities across three pretraining corpora, stratified by linguistic category (orthographic vs. vocabulary). Probabilities are derived from corpus-specific frequencies for 1,813 word pairs, representing mutually exclusive dialectal usage. All distributions show a consistent skew toward AmE variants, especially in spelling patterns. Additional corpora are shown in Appendix (Figure 6).

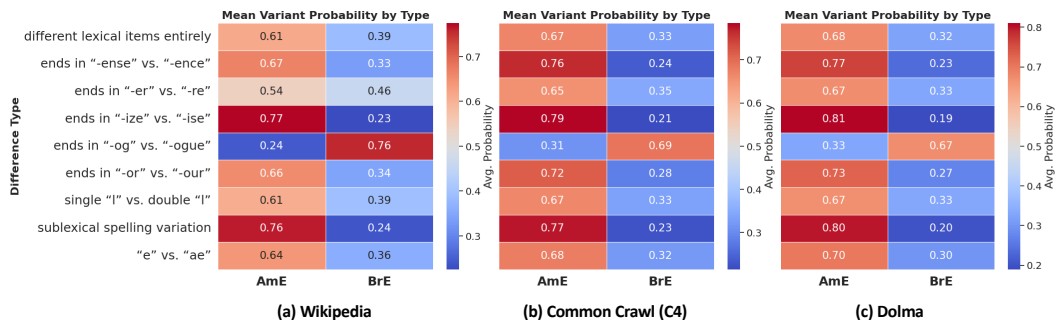

Figure 3: Average probability of observing AmE or BrE variants across word pairs, grouped by linguistic difference type and visualized for three pretraining corpora. Probabilities are computed by normalizing variant frequencies within each pair and averaging across each category, which includes orthographic and vocabulary-based differences. Each cell shows the mean probability for a variant type, with darker shades indicating stronger corpus-level preference. Results consistently reveal a skew toward American English. Additional corpora are presented in Appendix (Figure 7).

To further examine corpus-level dialectal skew, we analyze two complementary visualizations. Figure 2 presents violin plots of AmE vs. BrE variant probabilities stratified by linguistic category. These distributions reveal more pronounced skew for orthographic variants, which cluster toward AmE-preferred spellings. Vocabulary-based differences show slightly more balanced distributions, yet still lean toward AmE variants. Figure 3 further decomposes these trends across *ten* subcategories (e.g., `-ize` vs. `-ise`, `-og` vs. `-ogue`). While most categories reveal dominant AmE preference, one notable exception is the `-og` vs. `-ogue` group, where usage is comparatively balanced. This can be attributed to enduring usage of British spellings such as `catalogue` and `dialogue` in American academic and formal contexts (Neumann, 2023).

> **Key Takeaway:** These results empirically substantiate the presence of dialectal skew in foundational LLM corpora that may propagate into tokenization preferences and model outputs (RQ2 and RQ3).

## 6 RQ2: QUANTIFYING REPRESENTATION IN REGIONAL TOKENIZERS

Tokenization is a foundational yet underexamined component of the LLM pipeline (Ali et al., 2024), with potential to introduce dialectal skew before any model inference or generation occurs. This research question probes whether subword tokenizers, particularly those developed in diverse geopolitical contexts (e.g., USA, Europe, China, and postcolonial regions), encode American and British English variants with equal efficiency in practice.

We hypothesize that tokenizers may encode implicit dialectal preferences due to imbalances in pretraining corpora, vocabulary construction, or regional design goals. If AmE variants are encoded with fewer subword splits than BrE counterparts, it implies latent favoritism toward AmE forms, affecting fluency, latency, token budget, long-context handling, and lexical preferences (Petrov et al., 2023; Ahia et al., 2023), even when the underlying corpora are dialectally balanced.

Table 3: Fertility scores for AmE and BrE variants across a diverse set of tokenizers, segmented by orthographic and vocabulary-based differences. Lower fertility indicates more efficient tokenization. $\Delta_o$ and $\Delta_v$ represent the relative gap between AmE and BrE forms. Highlighted **bold** and underlined values denote the best and second-best results. Region-specific tokenizers show varying degrees of dialectal asymmetry. All differences are statistically significant with $p$-value $< 0.01$ based on the Wilcoxon signed-rank test, except for Velvet-2B in the orthographic category, marked with †.

| Tokenizers | Origin Country | Model Access | Vocab Size | Orthographic | | | Vocabulary | | |
|---|---|---|---|---|---|---|---|---|---|
| | | | | AmE (🇺🇸) | BrE (🇬🇧) | $\Delta_o$ | AmE (🇺🇸) | BrE (🇬🇧) | $\Delta_v$ |
| GPT-4 | USA (🇺🇸) | 🔒 | 100K | 2.73 | 2.86 | ↑ 4.76 % | 2.27 | 2.64 | ↑ 16.30 % |
| GPT-4o | USA (🇺🇸) | 🔒 | 200K | 2.65 | 2.77 | ↑ 4.53 % | 2.21 | 2.57 | ↑ 16.29 % |
| Llama-3.3-70B | USA (🇺🇸) | 🔓 | 128K | 2.72 | 2.85 | ↑ 4.78 % | 2.27 | 2.63 | ↑ 15.86 % |
| Gemma-3-27B | USA (🇺🇸) | 🔓 | 262K | **2.40** | **2.53** | ↑ 5.42 % | **2.02** | **2.35** | ↑ 16.34 % |
| DeepSeek-V3 | China (🇨🇳) | 🔓 | 128K | 2.71 | 2.80 | ↑ 3.32 % | 2.37 | 2.67 | ↑ 12.66 % |
| Mistral-Small-24B | France (🇫🇷) | 🔓 | 131K | 2.81 | 2.89 | ↑ 2.85 % | 2.45 | 2.79 | ↑ 13.88 % |
| StableLM-2-1.6B | UK (🇬🇧) | 🔓 | 100K | 2.73 | 2.86 | ↑ 4.76 % | 2.27 | 2.64 | ↑ 16.30 % |
| Velvet-2B | Italy (🇮🇹) | 🔓 | 127K | 2.90 | 2.88 | ↓ 0.69 %† | 2.40 | 2.72 | ↑ 13.33 % |
| Falcon3-7B | UAE (🇦🇪) | 🔓 | 131K | 2.44 | 2.56 | ↑ 4.92 % | 2.03 | 2.41 | ↑ 18.72 % |

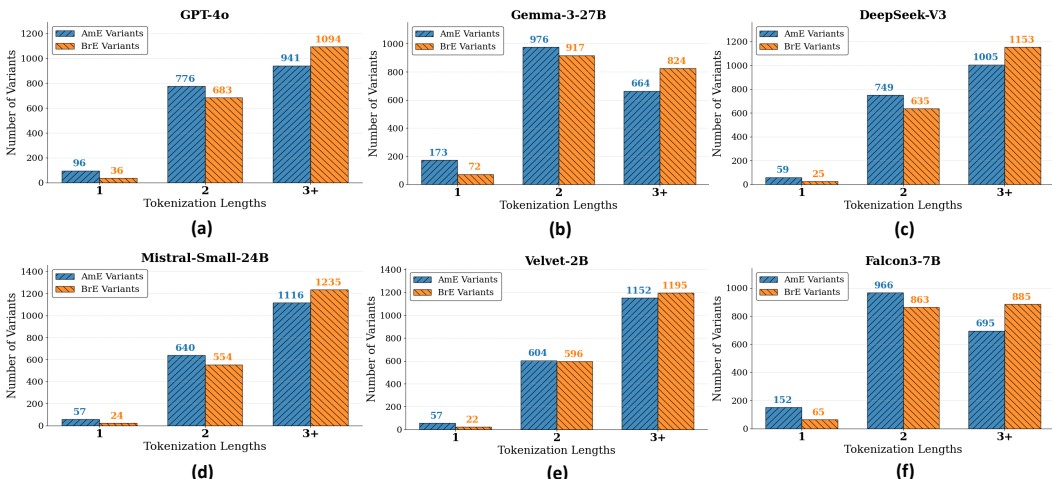

Figure 4: Granularity analysis of tokenization lengths for AmE and BrE variants across six tokenizers. Each subplot shows the count of variant pairs split into 1, 2, or 3+ subwords. BrE variants consistently exhibit more 3+ segmentations, indicating less efficient tokenization (other tokenizers in Figure 8).

**Methodology** To assess representational parity at the tokenization layer, we analyze how fairly publicly available regional tokenizers encode AmE–BrE lexical variants. We adopt *fertility*, defined as the average number of subword tokens per word, as our core diagnostic; following its widespread use in evaluating tokenization efficiency (Rust et al., 2021; Ahia et al., 2023; Ali et al., 2024). Lower fertility indicates higher encoding efficiency, while disparities in fertility between dialectal variants reflect representational asymmetries. Parity is achieved when fertility values are comparable across the AmE and BrE forms of each pair.

However, fertility captures only a mean-level view. To obtain a more granular picture, we compute the full *token-length distribution* for each tokenizer, reflecting how frequently words are split into 1, 2, 3, or more subword units. We refer to this distributional diagnostic as *granularity*. Unlike fertility, granularity reveals long-tail behavior, highlighting how often tokenizers produce excessive fragmentation, especially for dialect-specific forms. It also offers insight into how subword vocabularies allocate their finite capacity across dialects.

**Results & Analysis** Table 3 and Figure 4 reveal consistent asymmetries in how regional tokenizers encode dialectal variants. Across all models, British forms yield higher fertility (i.e., are tokenized into more subwords), than their American counterparts. This disparity is more pronounced for vocabulary-based differences (up to $\Delta_v = 18.72\%$). Orthographic differences show smaller but systematic gaps ($\Delta_o$ ranging from 2.85% to 5.42%).

Tokenizers developed outside the USA, particularly in Europe (Mistral, Velvet) and China (DeepSeek), exhibit improved BrE coverage. Velvet (Italy) uniquely favors BrE orthographic forms ($\Delta_o = -0.69\%$), while DeepSeek (China) shows the lowest vocabulary skew ($\Delta_v = 12.66\%$). DeepSeek also demonstrates a relatively balanced pattern across variants, which suggests the possibility of balanced exposure to dialects[2], or reflects potential differences in pretraining corpora. Gemma achieves the lowest overall fertility across both dialects due to its large vocabulary size (262K), suggesting that controlled vocabulary expansion, when guided by dialect-aware corpora, can improve overall dialectal representation. Granularity patterns in Figure 4 further corroborate these trends. BrE variants are consistently overrepresented in the 3+ token bin across tokenizers. Falcon and Gemma tokenize more compactly, reducing excessive fragmentation, especially in the long-tail bins. Notably, StableLM (UK) mirrors GPT-4 in its asymmetries (also fertility scores in Table 3) due to direct tokenizer reuse, illustrating the risks of transplanting tokenizers without regional adaptation.

> **Key Takeaway:** These findings expose a consistent yet underexplored layer of dialectal skew embedded within tokenizer design. They highlight the need for dialect-sensitive vocabulary allocation strategies and caution against blindly adopting pretrained tokenizers (Section 8).

# 7 RQ3: EVALUATING DIALECTAL PREFERENCES IN LLM GENERATION

Our goal is to evaluate dialectal preferences in LLM generations by assessing whether outputs align with AmE or BrE. To this end, we introduce DIALIGN, a novel scoring method that aims to capture commonly preferred lexical, grammatical, structural, stylistic, and multi-word contrasts. Given a question and a model-generated response, the objective is to *estimate* the dialectal alignment of the response rather than its factual correctness. DIALIGN is simple, dynamic, and training-free, leveraging distributional evidence and therefore applicable across diverse contexts, including pretraining data audits and the filtering of both existing corpora and synthetic data (Section 8).

## 7.1 DIALIGN: DIALECTAL ALIGNMENT SCORE

DIALIGN is a frequency–driven scoring function that *estimates* the alignment of a text toward AmE or BrE using historical corpus statistics. For a given input $x$, it computes $(P_{\text{AmE}}, P_{\text{BrE}})$, interpreted as alignment probabilities with $P_{\text{AmE}} + P_{\text{BrE}} = 1$. The procedure consists of *four* stages:

**n-gram Extraction.** We extract contiguous $n$-grams of input to capture grammatical, structural, stylistic, and multi-word contrasts. For a tokenized input $x = (t_1, \ldots, t_N)$, let $\mathcal{G}(x)$ denote all contiguous $n$-grams of length $2 \leq n \leq 5$:

$$\mathcal{G}(x) = \bigcup_{n=2}^{5} \{ g = (t_i, \ldots, t_{i+n-1}) \mid 1 \leq i \leq N - n + 1 \}.$$

To reduce topical and function-word artifacts, we discard any $g \in \mathcal{G}(x)$ that **(i)** contains a named entity (person, organization, location), or **(ii)** consists exclusively of stopwords.

**Frequency Lookup.** For each $g \in \mathcal{G}(x)$, we query the Google Books Ngram corpus[5] to obtain normalized average yearly frequencies $f_{\text{AmE}}(g)$ and $f_{\text{BrE}}(g)$ over a period $[y_{\min}, y_{\max}]$, reflecting forms mostly used or commonly preferred in AmE and BrE. If either frequency is zero, $g$ is discarded.

**Signed Divergence per n-gram.** Define the log-ratio

$$\text{LR}(g) = \log_2 \left( \frac{f_{\text{AmE}}(g)}{f_{\text{BrE}}(g)} \right).$$

Positive values indicate AmE preference, negative values BrE preference, and $\text{LR}(g) = 0$ indicates no dialectal signal. To down-weight ambiguous $n$-grams, we introduce a base divergence weight:

$$\delta(g) = \frac{|f_{\text{AmE}}(g) - f_{\text{BrE}}(g)|}{f_{\text{AmE}}(g) + f_{\text{BrE}}(g)} \in [0, 1).$$

---

[5] https://books.google.com/ngrams/, which provides frequency distributions from large-scale historical corpora, grouped into AmE and BrE, and capturing both canonical variants and broader structural contrasts.

Table 4: Dialectal preferences of LLMs on Natural Questions (*formal*) and ELI5 (*informal*) domains. We report percentages of AmE under default English and British English (en-GB) prompts, with mean AmE confidence scores in brackets. AmE is the dominant default, though non-U.S. models and informal domains show greater BrE uptake. **Bold** marks the lowest AmE percentage in each column; underlined marks the second-lowest.

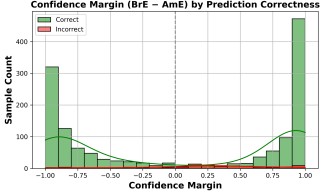

| LLMs | Origin Country | Natural Questions [formal] | | ELI5 [informal] | |
|---|---|---|---|---|---|
| | | Default English (AmE$^{\text{Default}}$) | British English (AmE$^{\text{BrE}}$) | Default English (AmE$^{\text{Default}}$) | British English (AmE$^{\text{BrE}}$) |
| GPT-4o | USA (🇺🇸) | 79.00% [0.81] | 45.33% [0.77] | 77.00% [0.82] | 34.67% [0.78] |
| Gemini-2.0-flash | USA (🇺🇸) | 76.00% [0.83] | 51.00% [0.79] | 75.33% [0.82] | 42.33% [0.80] |
| Claude-3.7-sonnet | USA (🇺🇸) | 75.67% [0.85] | 42.33% [0.80] | 73.33% [0.86] | 37.67% [0.78] |
| Llama-3.3-70B | USA (🇺🇸) | 74.67% [0.82] | 47.33% [0.78] | 69.00% [0.79] | **30.00%** [0.76] |
| Gemma-3-27B | USA (🇺🇸) | **69.33%** [0.81] | 45.67% [0.78] | 68.33% [0.83] | 38.00% [0.78] |
| DeepSeek-V3 | China (🇨🇳) | 74.67% [0.82] | **41.67%** [0.78] | 73.33% [0.84] | 40.47% [0.76] |
| Mistral-Small-24B | France (🇫🇷) | 73.67% [0.81] | 48.00% [0.75] | 72.33% [0.82] | 38.67% [0.76] |
| StableLM-2-1.6B | UK (🇬🇧) | 74.00% [0.79] | 69.67% [0.78] | 73.33% [0.77] | 67.00% [0.77] |
| Velvet-2B | Italy (🇮🇹) | 72.91% [0.80] | 69.33% [0.81] | 71.00% [0.80] | 67.67% [0.80] |
| Falcon3-7B | UAE (🇦🇪) | 73.67% [0.80] | 66.00% [0.79] | **66.00%** [0.81] | 63.33% [0.77] |

Figure 5: Meta-evaluation of DI-ALIGN. Performance is shown via confusion matrix, confidence margin distribution, and summary metrics (Acc, Precision, Recall, F1).

We then apply a lexicon-based boost using the variant lexicon $\mathcal{D}$ (Section 4):

$$w(g) = \begin{cases} \delta(g) \cdot \beta & \text{if } g \cap \mathcal{D} \neq \emptyset, \\ \delta(g) & \text{otherwise,} \end{cases}$$

where $\beta > 1$ is a boosting constant that favors dialect-diagnostic $n$-grams.

**Aggregation and Normalization.** Partition $\mathcal{G}(x)$ by the sign of $\text{LR}(g)$:

$$S_{\text{AmE}} = \sum_{\substack{g \in \mathcal{G}(x) \\ \text{LR}(g) > 0}} \text{LR}(g) \cdot w(g), \quad S_{\text{BrE}} = \sum_{\substack{g \in \mathcal{G}(x) \\ \text{LR}(g) < 0}} |\text{LR}(g)| \cdot w(g).$$

These are normalized to probabilities:

$$P_{\text{AmE}} = \frac{S_{\text{AmE}}}{S_{\text{AmE}} + S_{\text{BrE}}}, \quad P_{\text{BrE}} = \frac{S_{\text{BrE}}}{S_{\text{AmE}} + S_{\text{BrE}}}.$$

Finally, an input $x$ is classified by majority alignment:

$$\hat{y}(x) = \arg\max_{d \in \{\text{AmE}, \text{BrE}\}} P_d.$$

Full implementation details are given in Appendix C, and Appendix D provides an illustrative, step-by-step walkthrough with parallel passages highlighting contrasts in spelling, vocabulary, grammar, and style, reflecting forms mostly used or preferred in each variety.

**Meta-evaluation of DiAlign** To validate DIALIGN, we assembled 1,500 short news-style texts balanced across AmE and BrE (750 each), drawn from HuffPost U.S. News and BBC England sources (see Appendix E for details). On this benchmark, DIALIGN achieves 93.2% accuracy, 90.7% precision, 96.3% recall, and an F1 score of 93.4, as shown in Figure 5. An ablation study (Table 5) shows that divergence weighting and boosting provide complementary gains, with the largest drop observed when both are removed. The confidence margin distribution in Figure 5 indicates that correct predictions are mostly made with high certainty, while errors cluster near the decision boundary.

**Experimental Setup** We assess dialectal preferences in open-domain QA across two registers: *formal* (Natural Questions (NQ) (Kwiatkowski et al., 2019)) and *informal* (ELI5 (Fan et al., 2019)). To avoid lexical priming, we discard questions containing any AmE or BrE variants. For each question, we elicit two generations under the language conditions *English* and *British English (en-GB)*. Alignment is then estimated with DIALIGN, which produces $(P_{\text{AmE}}, P_{\text{BrE}})$ and assigns the dialect via arg max (see Section 7.1). We report both the percentages of AmE-classified generations and the mean AmE alignment confidence $P_{\text{AmE}}$ (*shown in brackets*) in Table 4, denoted AmE$^{\text{Default}}$ for the English prompt and AmE$^{\text{BrE}}$ for the British English control. Full details of the datasets, filtering, prompt template (Figure 11), model list, and decoding parameters are provided in Appendix F.

**Results & Analysis** As shown in Table 4, AmE is the dominant generative default. Under the `default English` condition, most models produce 65–80% AmE outputs often with high confidence ($> 0.80$). Even when explicitly prompted with `British English (en-GB)`, AmE persists, rarely dropping below 40%. U.S.-developed models show the strongest AmE preferences, while non-U.S. models shift slightly more toward BrE, reflecting the influence of pretraining corpora and tokenizer design [see RQ1 (§5) and RQ2 (§6)]. Notably, Gemma achieves relatively higher BrE alignment, likely aided by its large 262K vocabulary, as discussed in RQ2 (§6).

Dialectal skew also varies by domain. In NQ (formal/encyclopedic), AmE dominates, with BrE prompts producing only partial shifts. In contrast, ELI5 (informal/conversational) shows greater BrE uptake, with models like LLaMA-3 dropping to ∼30% AmE under `en-GB`, likely reflecting its social media–oriented training data. This indicates that conversational registers provide more lexical flexibility, whereas formal contexts reinforce standardized AmE norms embedded in pretraining data [RQ1 (§5)]. The persistence of AmE even under BrE prompting aligns with the hypothesis of a latent English subspace (Wendler et al., 2024; Zhao et al., 2024); our findings suggest this subspace is structurally AmE dominant, creating a gravitational pull that resists surface-level dialectal steering.

> **Key Takeaway:** AmE is the entrenched generative default across LLMs, persisting even under BrE prompts. BrE uptake is stronger in informal domains but limited in formal ones, revealing *structural biases* shaped jointly by pretraining data [RQ1 (§5)] and tokenizer design [RQ2 (§6)]. This raises inclusivity concerns, as users expecting BrE norms (e.g., in education, journalism, or institutional contexts) may encounter outputs subtly misaligned with their linguistic expectations.

## 8 DISCUSSION & RECOMMENDATIONS

**Dialectal Skew and Broader Implications.** The dialectal skew observed in LLMs likely extends beyond linguistic variation, reflecting how pretraining data can embed broader cultural tendencies. By privileging AmE, models may carry forward its norms, values, and worldviews; shaping which knowledge is legitimized and which practices are marginalized. This resonates with broader critiques that LLMs can amplify hegemonic perspectives encoded in training corpora (Bender et al., 2021). Such patterns suggest that dialectal bias intersects with epistemic and political asymmetries, raising important considerations for technical AI governance and Sovereign AI initiatives (Reuel et al., 2025).

**Balancing Pretraining Data for Improved Dialectal Representation.** Dialectal skew is partly rooted in the construction of pretraining corpora [RQ1 (§5)]. Large web-scale datasets such as Common Crawl (C4) and Dolma (Figure 10) often include metadata such as source URLs, which can be leveraged to enrich dialectal coverage for World Englishes that build on BrE, such as Canadian or Indian English (e.g., `.ca`, `.in`). For instance, BrE coverage can be increased by selectively sampling from `.uk` domains. When naturally occurring data are scarce, synthetic data generation may be considered (Liu et al., 2024b); however, such generations risk defaulting to AmE. In this setting, DIALIGN provides a safeguard by verifying whether synthetic samples align with BrE before inclusion (§7.1), thereby supporting balanced and representative corpus design. These pretraining data can be used to continue pretraining base models, improving dialectal representation in LLMs.

**Dialect-Sensitive Tokenizer Design.** Another source of bias arises from reusing existing tokenizers without addressing dialectal asymmetries [RQ2 (§6)]. Current vocabularies disproportionately favor AmE variants, structurally skewing generation. A practical remedy is dialect-sensitive vocabulary extension: using our AmE–BrE lexicon and granularity-based diagnostics to identify BrE tokens absent from the base tokenizer and injecting them via controlled vocabulary expansion (Tejaswi et al., 2024). *We acknowledge and discuss the study's limitations and some future directions in Appendix J.*

## 9 CONCLUSION

This paper presents the first systematic audit of dialectal asymmetries across the LLM development pipeline. By triangulating evidence from pretraining corpora, tokenizer behavior, and generative outputs, we show that AmE emerges as the default and BrE is consistently disadvantaged, revealing how digital dominance manifests as structural bias. Interpreted through a holistic postcolonial lens, these findings highlight risks of linguistic homogenization and epistemic injustice, and motivate balanced corpora, dialect-sensitive tokenizers, and alignment for inclusive language technologies.

## ETHICS STATEMENT

This work examines dialectal asymmetries in LLMs, focusing on American and British English through a postcolonial lens. It does not involve human subjects, personal data, or sensitive attributes. All datasets analyzed are publicly available corpora (e.g., Common Crawl, Wikipedia; Appendix G) and were used solely for research purposes. The curated AmE–BrE lexicon was derived from publicly accessible sources (Table 7) and will be released for non-commercial research under a CC BY-NC-SA 4.0 license[6], containing no personal or proprietary material.

While our analysis is explicitly restricted to English, we acknowledge that "wordhood" is a language-specific construct and that many languages lack clear orthographic word boundaries or segment linguistic units in very different ways. We therefore view this work as an English-specific instantiation of our framework; extensions to other languages will need to adapt segmentation assumptions to local linguistic norms rather than imposing a Western-centric notion of words.

The ethical relevance of this research lies in documenting and quantifying structural linguistic biases that privilege American English as the de facto norm in LLM development. Such biases risk perpetuating epistemic injustice and linguistic homogenization in global AI deployment. Our aim is constructive: by exposing these asymmetries, we provide tools (e.g., DIALIGN) and evidence to support more inclusive, transparent, and dialect-aware language technologies. No harmful applications are proposed, and all methodological artifacts were designed for responsible auditing. In constructing DIALIGN, we explicitly excluded named entities (e.g., personal names, organizations, and locations) to avoid privacy or reputational risks. Also, limitations of our study are discussed in Appendix J.

## REPRODUCIBILITY STATEMENT

We have made substantial efforts to ensure the reproducibility of our results. The sources used for curating the AmE–BrE lexicon are presented in Table 7, while the typology of variants and the classification scheme are documented in Appendix B and are also shared in the supplementary material. All six pretraining corpora analyzed are publicly available, with references and HuggingFace links provided in Appendix G, together with a detailed description of our preprocessing pipeline in Appendix I. The implementation details of DIALIGN, including meta-evaluation procedures, are presented in Appendix C and Appendix E, and the experimental setup for assessing dialectal preferences in LLM generation is provided in Appendix F. The code, preprocessing scripts, resources, and test samples are all included in the supplementary material, with a cleaner release version to be made available upon paper acceptance.

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

# Supplementary Material: Appendices

## A  BRIEF HISTORY

The divergence between British and American English is the outcome of both deliberate acts of standardization and broader sociopolitical forces. A formalized "British standard" crystallized with Samuel Johnson's 1755 dictionary, which codified spelling conventions and consolidated authority in literary and educational practice. In contrast, an "American standard" emerged with Noah Webster's 1828 dictionary, which advocated simplified and distinct spellings as a marker of cultural independence from Britain (Baker, 2017). These codifications established the orthographic contrasts that remain central to dialectal variation today.

The global dissemination of English was inseparable from British colonial expansion. Across Africa, Asia, the Caribbean, and the Pacific, British English became entrenched in governance, education, and law, often persisting as the official or de facto standard after independence. Today, it continues to hold normative prestige across much of the Commonwealth, underpins European Union institutions, and is actively promoted by the UK through initiatives such as the Oxford Dictionary, the British Council, and standardized assessments like IELTS. This trajectory is briefly illustrated in Figure 1, which depicts the mid-twentieth-century wave of decolonization, when many newly sovereign nations retained the linguistic imprint of British English in state institutions.

By contrast, American English spread primarily through twentieth-century cultural and economic influence, propelled by mass media, technological innovation, and global commerce (Crystal, 2003; Nordquist, 2024). It dominates digital communication and popular culture, positioning AmE as a de facto global norm. Importantly, the authority of both standards rests not on linguistic merit but on sociopolitical power and institutional reinforcement (Milroy & Milroy, 1999; Lippi-Green, 2012). This layered history explains why AmE and BrE continue to exert cultural and normative influence in different regions and underscores the sociolinguistic significance of examining dialectal alignment in modern foundation models for global inclusivity.

## B  DIALECTAL VARIANT GROUPING

To structure our set of 1,813 AmE–BrE word-variant pairs, we employ a deterministic, rule-based procedure that assigns each pair to exactly one of ten mutually exclusive groups, in descending order of precedence. These ten groups are further collapsed into three high-level categories: *Orthographic/Spelling*, *Vocabulary*, and *Uncategorized*.

**Group Definitions.**  We classify each pair according to the first matching rule in the following list:

- **Group 1 (-or vs. -our):** suffix -or (AmE) $\leftrightarrow$ suffix -our (BrE).
- **Group 2 (-ize vs. -ise):** suffix -ize (AmE) $\leftrightarrow$ suffix -ise (BrE).
- **Group 3 (-er vs. -re):** suffix -er (AmE) $\leftrightarrow$ suffix -re (BrE).
- **Group 4 (-og vs. -ogue):** suffix -og (AmE) $\leftrightarrow$ suffix -ogue (BrE).
- **Group 5 (single "l" vs. double "ll"):** AmE single "l" $\leftrightarrow$ BrE double "ll".
- **Group 6 (-ense vs. -ence):** suffix -ense (AmE) $\leftrightarrow$ suffix -ence (BrE).
- **Group 7 (ae vs. e):** BrE form contains "ae" where the AmE form replaces it with "e".
- **Group 8 (same length, small edit):** pairs of equal length whose Levenshtein distance is 1–2 (i.e., minor sublexical shifts).
- **Group 9 (different words):** pairs whose lengths differ or whose edit distance exceeds 2 (i.e., entirely distinct lexical items).
- **Group 10 (miscellaneous):** all remaining pairs not captured by the above rules.

**Category Assignment.**  We map each of the ten groups to one of three overarching categories:

- **Orthographic/Spelling (Groups 1–8):** These groups reflect systematic spelling alternations (e.g. "–or"/"–our", "–ize"/"–ise", double vs. single "l", "ae" vs. "e", etc.).
- **Vocabulary (Group 9):** True lexical substitutions (e.g. "elevator" vs. "lift") in which the AmE and BrE forms share no orthographic root.

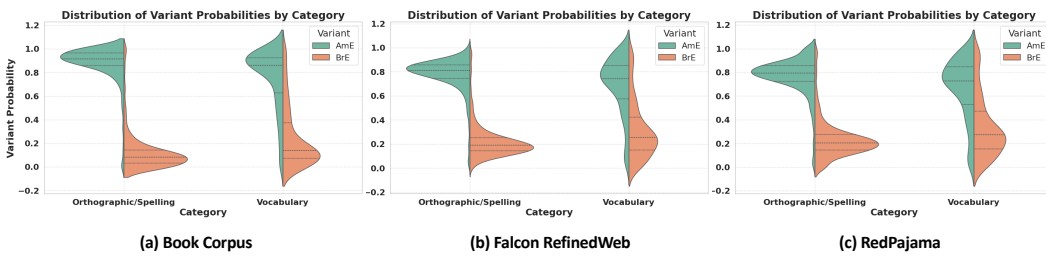

Figure 6: Violin plots showing the distribution of AmE vs. BrE variant probabilities across three pretraining corpora **(a)** Book Corpus, **(b)** Falcon RefinedWeb, and **(c)** RedPajama, stratified by linguistic category (orthographic vs. vocabulary). Probabilities are derived from corpus-specific frequencies for 1,813 word pairs, representing mutually exclusive dialectal usage. All distributions show a consistent skew toward AmE variants, especially in spelling patterns.

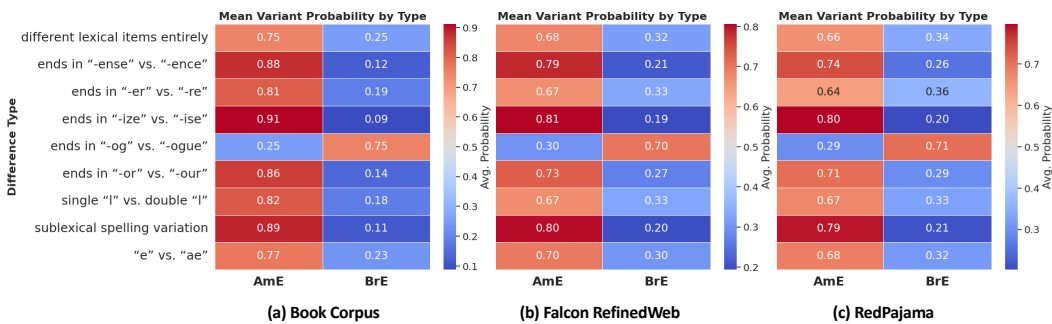

Figure 7: Average probability of observing AmE or BrE variants across word pairs, grouped by linguistic difference type and visualized for three pretraining corpora: **(a)** Book Corpus, **(b)** Falcon RefinedWeb, and **(c)** RedPajama. Probabilities are computed by normalizing variant frequencies within each pair and averaging across each category, which includes orthographic and vocabulary-based differences. Each cell shows the mean probability for a variant type, with darker shades indicating stronger corpus-level preference. Results consistently reveal a skew toward AmE.

- **Uncategorized (Group 10):** Exceptional or edge-case pairs that do not fit any of the above patterns.

This classification scheme is both exhaustive and mutually exclusive, ensuring robust coverage of our curated variant inventory. It provides a linguistically principled basis for analyzing American English (AmE) vs. British English (BrE) variants.

## C  IMPLEMENTATION DETAILS OF DIALIGN

We provide here the implementation details of the DIALIGN scoring procedure used to estimate American and British English alignment in model generations. The design emphasizes efficiency and robustness.

**Parameterization.** The key parameters of DIALIGN are as follows:

- **$n$-gram range:** $n \in \{2, 3, 4, 5\}$, enabling the capture of grammatical, structural, multi-word contrasts, and stylistic variation beyond isolated tokens while avoiding sparsity at higher orders. This range aligns with the Google Books Ngram corpus, which provides reliable statistics up to 5-grams.

- **Temporal range:** $[y_{\min}, y_{\max}] = [1950, 2022]$, balancing contemporary representativeness with sufficient historical depth to smooth short-term fluctuations.

- **Smoothing:** set to 0 to use raw frequency distributions. In practice, unsmoothed counts yield clearer discriminative signals for dialectal contrasts.

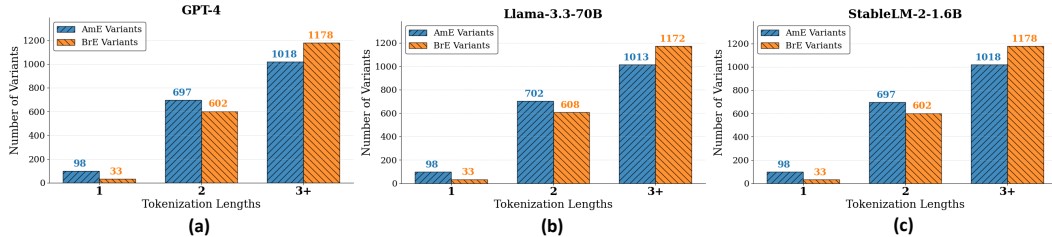

Figure 8: Granularity analysis of tokenization lengths for AmE and BrE variants across three regional tokenizers: **(a)** GPT-4 **(b)** Llama-3.3-70B, and **(c)** StableLM-2-1.6B. Each subplot shows the count of variant pairs split into 1, 2, or 3+ subwords. BrE variants consistently exhibit more 3+ segmentations, indicating finer-grained and less efficient tokenization.

- **Boosting factor:** $\beta = 1.5$, applied to lexicon-derived dialectal markers to amplify their influence in the alignment score.

While these parameter choices are principled, they are not unique. Alternative configurations of $n$-gram order, temporal window, smoothing, or boosting may yield different or improved performance. The present setup serves as a transparent, reproducible baseline that future work can refine or extend.

**Frequency Lookup.** Frequencies are collected dynamically via the Google Books Ngram API using the `requests` library, avoiding the need to download terabytes of raw corpus data (Lin et al., 2012), which is impractical in most academic settings. For each candidate $n$-gram, we query both American English (AmE, corpus ID 17) and British English (BrE, corpus ID 6), aggregating case-insensitive counts. The API returns normalized yearly frequencies (relative to the total number of tokens per year), which we average over the specified range, with exponential-backoff retries to guard against transient failures. To avoid repeated calls, we maintain a persistent $n$-gram cache on disk: once an $(n\text{-gram}, \text{corpus})$ pair has been queried, subsequent samples reuse the cached value. This setup yields an online, efficient, and reproducible mechanism for alignment estimation.

**Filtering.** To reduce topical and functional noise, $n$-grams are excluded if they:

- contain named entities such as persons, organizations, or locations (e.g., "Barack Obama", "New York"), detected using NLTK's named entity recognition (NER) via chunking[7], or
- consist solely of stopwords (e.g., "in the", "and a"), identified using the NLTK stopword list.

This filtering step ensures that retained $n$-grams are stylistically and grammatically informative.

Overall, this procedure yields alignment scores that reflect grammatical and stylistic choices at the $n$-gram level while integrating informative priors from lexicon-based boosting. The design choices align with the broader methodological goals of capturing structural dialectal skew.

## D WALKTHROUGH OF DIALIGN WITH ILLUSTRATIVE INPUT

To illustrate the operation of DIALIGN, we provide parallel input texts in American English (AmE) and British English (BrE). Figure 9 shows the two versions of the same passage, highlighting contrasts in orthography, vocabulary, syntax, and style. DIALIGN first segments the input into contiguous $n$-grams ($n = 2 \ldots 5$), then in the frequency lookup stage queries Google Books N-grams (AmE corpus ID 17, BrE corpus ID 6) to obtain corpus frequencies. The aggregated evidence is finally normalized to return alignment probabilities $(P_{\text{AmE}}, P_{\text{BrE}})$.

The passages embed a wide spectrum of dialectal contrasts, reflecting forms that are mostly used or commonly preferred in one variety over the other:

- **Spelling:** *traveler (AmE) / traveller (BrE), organizing (AmE) / organising (BrE), realized (AmE) / realised (BrE), program (AmE) / programme (BrE), spilled (AmE) / spilt (BrE).*

---

[7]NLTK provides implementations for NER and stopword lists; see https://www.nltk.org

"The *traveler* *was* *organizing* notes in the *lecture hall on the weekend, and he realized he just ate,* so he wasn't hungry. *The team is winning, right?* He *stood in line for a cookie, learned* the result from the *program, and dreamed* of finishing by *December 31, 2024. They suggested he go,* though *he must have already done* the work. *At the train station,* he spoke of having *spilled* his tea before *taking the elevator* to the *first floor.* Later, his colleague said she had gotten a new book while in college, ordered French fries with a side of ketchup, wore a sweater over her shirt, and bought potato chips at the grocery store. In the fall semester she lived on Main Street near the gas station, took a math class in the parking lot building, and always carried her cell phone in her purse."

**American English (AmE)**

"The traveller was organising notes in the lecture theatre at the weekend, and he realised he had just eaten, so he wasn't hungry. The team are winning, aren't they? He queued for a biscuit, learnt the result from the programme, and dreamt of finishing by 31 December 2024. They suggested he should go, though he must have done the work already. At the railway station, he spoke of having spilt his tea before taking the lift to the ground floor. Later, his colleague said she had got a new book while at university, ordered chips with a side of tomato sauce, wore a jumper over her shirt, and bought crisps at the supermarket. In the autumn term she lived in High Street near the petrol station, took a maths course in the car park building, and always carried her mobile phone in her handbag.""

**British English (BrE)**

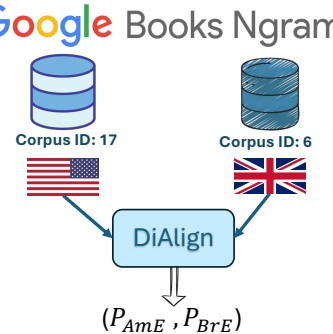

$(P_{AmE}, P_{BrE})$

Figure 9: Illustrative walkthrough of DIALIGN. Parallel passages in AmE and BrE highlight contrasts in spelling, vocabulary, grammar, and style, reflecting forms mostly used or preferred in each variety. Frequencies are retrieved from the Google Books Ngram corpora (AmE: ID 17, BrE: ID 6), and DIALIGN outputs alignment probabilities $(P_{AmE}, P_{BrE})$.

- **Vocabulary:** *cookie (AmE) / biscuit (BrE), elevator (AmE) / lift (BrE), lecture hall (AmE) / lecture theatre (BrE), train station (AmE) / railway station (BrE).*
- **Verb morphology (past tense):** forms such as *dreamed (AmE) / dreamt (BrE), learned (AmE) / learnt (BrE), gotten (AmE) / got (BrE).*
- **Tense and aspect:** *I just ate (AmE, simple past) / I've just eaten (BrE, present perfect).*
- **Collective noun agreement:** *The team is winning (AmE) / The team are winning (BrE).*
- **Discourse markers:** *right? (AmE) / aren't they? (BrE).*
- **Subjunctive usage:** *They suggested he go (AmE) / They suggested he should go (BrE).*
- **Auxiliary phrasing:** *must have already done (AmE) / must have done (BrE).*
- **Prepositional usage:** *on the weekend (AmE) / at the weekend (BrE).*
- **Date format:** *December 31, 2024 (AmE) / 31 December 2024 (BrE).*
- **Floor reference:** *first floor (AmE) / ground floor (BrE).*
- **Institutional idioms:** *in college (AmE) / at university (BrE), fall semester (AmE) / autumn term (BrE).*
- **Food collocations:** *French fries with a side of ketchup (AmE) / chips with a side of tomato sauce (BrE); potato chips at the grocery store (AmE) / crisps at the supermarket (BrE).*
- **Clothing collocations:** *wore a sweater (AmE) / wore a jumper (BrE).*
- **Transport and location idioms:** *on Main Street near the gas station (AmE) / in High Street near the petrol station (BrE); parking lot (AmE) / car park (BrE).*
- **Education phrases:** *took a math class (AmE) / took a maths course (BrE).*
- **Everyday objects:** *cell phone in her purse (AmE) / mobile phone in her handbag (BrE).*

By embedding orthographic, lexical, grammatical, and multi-word collocational contrasts in a unified passage, this walkthrough illustrates how DIALIGN leverages $n$-gram frequency divergences across $n = 2 \ldots 5$ to capture dialectal alignment. The example highlights that the method accounts not only for single-word substitutions but also for structural and idiomatic usage patterns.

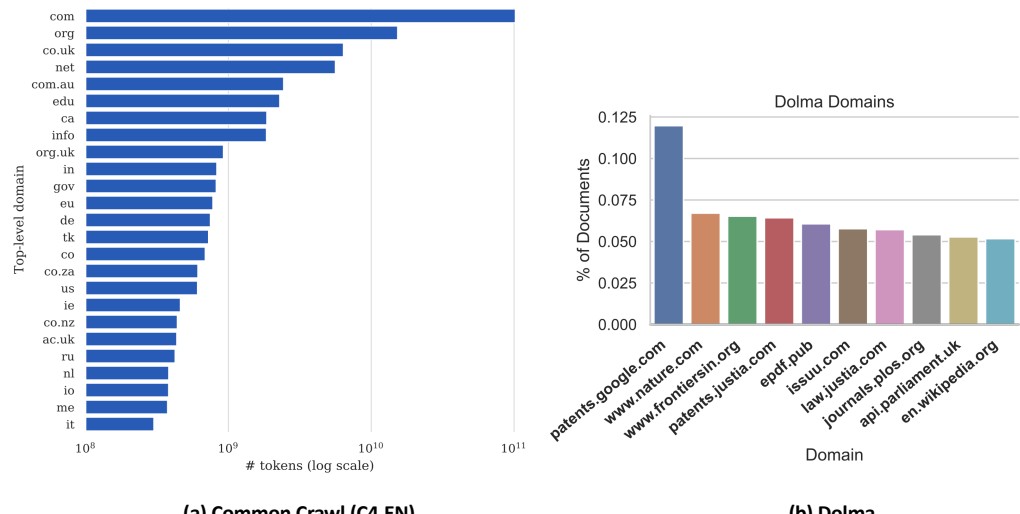

**(a) Common Crawl (C4.EN)**                    **(b) Dolma**

Figure 10: Domain distributions in two widely used pretraining corpora for LLMs. **(a)** Top-level domains in Common Crawl (C4.EN) (Raffel et al., 2020), showing heavy concentration in .com and .org with much lower representation of .co.uk, suggesting a potential AmE skew. **(b)** High-frequency domains in Dolma (Soldaini et al., 2024), reflecting a narrower, curated set of sources that is comparatively more balanced but remains predominantly U.S.-centric.

## E    DETAILS OF THE META-EVALUATION OF DIALIGN

To validate DIALIGN, we require texts that predominantly reflect American English (AmE) or British English (BrE) spelling, vocabulary, grammar, and other stylistic preferences. Since no standard dataset exists with explicit AmE–BrE annotations, we identified corpora where dialectal variation is strongly embedded in the source of the texts. These corpora serve as a reasonable proxy for meta-evaluating DIALIGN.

**BrE Samples.**    For BrE, we draw from the XL-Sum dataset (Hasan et al., 2021), which contains abstractive summaries across multiple languages sourced from the BBC News website[8]. BBC is a UK-based outlet that predominantly adopts British spelling and stylistic conventions, making it an appropriate source for BrE texts. We focus on the English portion of the dataset. Each data entry includes an id field indicating the article identifier; we select those beginning with the prefix uk-england- to ensure regional specificity. From these entries, we use the summary field as our text sample and randomly sample 750 instances.

**AmE Samples.**    For AmE, we use the News Category Dataset (Misra, 2022), which consists of headlines and short descriptions collected from HuffPost[9] across multiple topical categories. As HuffPost is a U.S.-based news outlet, it predominantly employs American spelling and usage. We specifically extract texts from the category "U.S. NEWS"[10]. For each entry, we take the short_description field and randomly select 750 instances.

**Dataset Statistics.**    This yields two balanced sets of 750 samples each, one from AmE sources and one from BrE sources, for a total of 1,500 samples. The average length of AmE samples is 34.4 words, while BrE samples average 24.7 words. The balanced design ensures comparability between the two groups while reflecting real-world stylistic preferences in their respective dialects.

**Justification and Limitations.**    Although these datasets are not explicitly annotated for dialect, the provenance of the sources (HuffPost for AmE, BBC for BrE) provides strong dialectal signals in

---

[8]https://huggingface.co/datasets/csebuetnlp/xlsum

[9]https://www.huffpost.com/

[10]https://huggingface.co/datasets/heegyu/news-category-dataset

Table 5: Ablation study of DIALIGN. We report classification performance (Accuracy, Precision, Recall, F1 Score) and average confidence (Avg. Conf.) for AmE and BrE predictions. Removing either the divergence weighting (DW) or boosting factor (BF) degrades performance, with the largest drop when both are removed.

| Ablations | Accuracy | Precision | Recall | F1 Score | Avg. Conf. (AmE) | Avg. Conf. (BrE) |
|---|---|---|---|---|---|---|
| DIALIGN (final) | 93.18 | 90.67 | 96.25 | 93.38 | 0.84 | 0.91 |
| – w/o Divergence Weight (DW) | 92.25 | 89.29 | 95.98 | 92.52 | 0.77 | 0.85 |
| – w/o Boosting Factor (BF) | 92.25 | 89.29 | 95.98 | 92.52 | 0.82 | 0.89 |
| – w/o Both (DW + BF) | 91.31 | 88.13 | 95.45 | 91.65 | 0.75 | 0.83 |

---

**Prompt template for RQ3: Evaluating Dialectal Preferences in LLM Generation**

```
Answer the following question in {language}. Write a single, coherent
paragraph in plain text, using descriptive and open-ended language. Avoid
bullet points, lists, or formatting.  Your response must be exactly
{WORD_LIMIT} words long—no more, no fewer. Count your words carefully.

Question: {question}
```

Figure 11: Prompt used to elicit model outputs under two language conditions. We set WORD_LIMIT= 50 and vary {language} $\in$ {English, British English (en-GB)}.

spelling, vocabulary, and grammatical constructions. Using such domain-specific proxies allows us to meta-evaluate DIALIGN in the absence of manually curated dialectal benchmarks. A limitation of this approach is that domain effects—such as differences in journalistic style between BBC and HuffPost—may introduce secondary variation beyond dialect. Nevertheless, the strong and systematic orthographic, lexical, and grammatical signals in these sources make them reliable proxies for AmE and BrE in our evaluation.

## F  DETAILS OF THE EXPERIMENTAL SETUP FOR RQ3

**Task and Objective.**   We evaluate dialectal preferences of LLM generations in an open-domain QA setting. Given a question, a model produces one short paragraph; the goal is to *estimate* the dialectal alignment of the output (AmE vs. BrE), not its factual correctness. Alignment is measured with DIALIGN, which outputs $(P_{\mathrm{AmE}}, P_{\mathrm{BrE}})$ and assigns the dialect via $\arg\max$ (see Section 7.1).

**Datasets.**   We use two complementary QA corpora to span formal and informal registers:

- **Natural Questions (NQ)** (Kwiatkowski et al., 2019)[11]: real Google search questions paired with Wikipedia answers; emphasizes formal, encyclopedic style.
- **ELI5** (Fan et al., 2019)[12]: community QA from Reddit; answers are conversational and descriptive; emphasizes everyday style.

Together these provide a broad stylistic spectrum (formal + informal) for testing dialectal defaults.

**Preprocessing and Sampling.**   To mitigate noise and reduce lexical leakage from prompts, we apply two filters:

- **Length filter:** discard items with fewer than 5 words in the question or fewer than 30 words in the gold answer. This ensures sufficient content for $n=2\dots 5$ scoring and aligns with our word-length constraint.

---

[11]https://huggingface.co/datasets/sentence-transformers/natural-questions
[12]https://huggingface.co/datasets/sentence-transformers/eli5

- **Variant-free questions:** remove questions containing any AmE or BrE lexical variants, using the full dialectal variant corpus of 3,626 entries (see Section 4), to avoid priming the output dialect.

From the filtered pool, we uniformly sample 600 questions (300 from NQ and 300 from ELI5).

**Prompting Protocol.** Each question is posed under two language settings: *English* and *British English (en-GB)*. The latter tests whether explicit British conditioning attenuates AmE defaults. We fix WORD_LIMIT= 50 to standardize length across models and datasets while providing enough context for bi- to 5-grams used by DIALIGN. The exact prompt is shown in Figure 11.

**Models and Decoding Parameters.** We evaluate a range of open- and closed-source LLMs spanning diverse geopolitical contexts (e.g., USA, Europe, China, UAE). To isolate prompt effects, decoding is held constant across both language conditions:

$$\text{temperature} = 0.0, \quad \text{top\_p} = 1.0, \quad \text{max\_tokens} = 512.$$

These settings enable clean comparison of dialectal tendencies: temperature=0 (greedy) removes sampling variance; top_p=1 disables nucleus filtering, keeping the model's *full vocabulary and grammar* available under both "English" and "British English (en-GB)" prompts; and max_tokens=512 prevents truncation of the 50-word target. With decoding fixed, any change in DiAlign scores is attributable to the prompt's dialectal conditioning rather than decoding noise or capacity limits.

**Scoring with DIALIGN.** Each generated paragraph is segmented into contiguous $n$-grams ($n=2\ldots5$). We query Google Books N-grams (AmE: ID 17, BrE: ID 6) for *normalized* yearly frequencies and aggregate evidence using signed log-ratios with bounded divergence weighting and a lexicon-based boost (see Section 7.1). This yields $(P_{\text{AmE}}, P_{\text{BrE}})$ and a predicted dialect via $\arg\max$.

**Zero-Signal Exclusions.** If both probabilities are zero, i.e., $(P_{\text{AmE}}, P_{\text{BrE}}) = (0, 0)$, we exclude the item from summary statistics.[13] Exclusion keeps reported rates focused on texts with measurable evidence.

**Outcome Measures.** For each dataset and language condition, we report (i) the percentage of generations classified as AmE and (ii) the mean AmE alignment confidence to visualize decision certainty. Our goal is to measure the default dialect of LLMs and, when prompted with British English (en-GB), determine how much of the output still aligns with AmE.

## G    DETAILS OF PRETRAINING DATASETS

We audited six widely used pretraining corpora to assess dialectal skew between American and British English (see Table 2). Below we briefly describe each dataset.

- **BookCorpus** (Zhu et al., 2015): A collection of unpublished novels widely used in NLP pre-training. It provides narrative-style English text, with approximately 74 million documents and 1.28 billion tokens.[14]
- **Wikipedia** (Foundation, 2024): Encyclopedic text from Wikipedia dumps, spanning diverse domains with formal writing style. The version used contains 6.4 million documents and 4.3 billion tokens.[15]
- **Common Crawl (C4)** (Raffel et al., 2020): A cleaned and deduplicated subset of Common Crawl web pages, containing large-scale web text used in many LLMs. It includes 365 million documents and 156 billion tokens.[16]

---

[13]This case is empirically rare and arises when surviving $n$-grams lack reliable corpus evidence in both dialects or when their weighted contributions cancel under divergence weighting, indicating insufficient dialectal signal.

[14]https://huggingface.co/datasets/bookcorpus/bookcorpus

[15]https://huggingface.co/datasets/wikimedia/wikipedia

[16]https://huggingface.co/datasets/allenai/c4

Table 6: Word-length adherence across models in RQ3: Evaluating Dialectal Preferences in LLM Generation (Section 7). Each model was instructed to produce exactly 50 words per answer. The table reports average length, standard deviation, and range across Natural Questions (formal) and ELI5 (informal), under both default English and British English (en-GB) prompts. Closed-source models (🔒) generally stay close to the target, while open-weight models (🔓) exhibit larger variance.

| | | Natural Questions (NQ) [formal] | | | | ELI5 [informal] | | | |
|---|---|---|---|---|---|---|---|---|---|
| | | Default English ($\%AmE^{Default}$) | | British English ($\%AmE^{BrE}$) | | Default English ($\%AmE^{Default}$) | | British English ($\%AmE^{BrE}$) | |
| LLMs | Model Access | #Words Avg. [SD] | Range [min-max] | #Words Avg. [SD] | Range [min-max] | #Words Avg. [SD] | Range [min-max] | #Words Avg. [SD] | Range [min-max] |
| GPT-4o | 🔒 | 50.19 [1.75] | [46–57] | 50.32 [1.61] | [45–56] | 50.41 [1.60] | [47–55] | 50.35 [1.53] | [46–55] |
| Gemini-2.0-flash | 🔒 | 52.74 [2.71] | [46–60] | 51.92 [2.89] | [45–61] | 53.10 [2.62] | [47–61] | 52.82 [2.95] | [44–60] |
| Claude-3.7-sonnet | 🔒 | 45.25 [2.29] | [39–52] | 45.49 [2.35] | [39–57] | 47.70 [2.45] | [42–56] | 47.38 [2.32] | [41–54] |
| Llama-3.3-70B | 🔓 | 41.68 [7.12] | [16–50] | 41.47 [7.44] | [18–50] | 41.33 [7.83] | [21–50] | 40.10 [8.48] | [18–51] |
| Gemma-3-27B | 🔓 | 49.03 [2.82] | [42–58] | 49.19 [2.64] | [44–61] | 49.18 [2.84] | [43–60] | 49.90 [2.74] | [44–58] |
| DeepSeek-V3 | 🔓 | 53.03 [4.13] | [48–102] | 53.40 [5.10] | [47–103] | 53.60 [4.66] | [48–98] | 53.33 [4.97] | [4–96] |
| Mistral-Small-24B | 🔓 | 51.44 [18.80] | [13–318] | 50.18 [10.97] | [20–88] | 57.26 [11.68] | [23–120] | 56.78 [10.43] | [34–101] |
| StableLM-2-1.6B | 🔓 | 82.94 [39.90] | [8–364] | 74.33 [40.15] | [8–351] | 96.82 [24.23] | [23–199] | 92.64 [24.76] | [41–199] |
| Velvet-2B | 🔓 | 47.07 [30.94] | [8–406] | 44.56 [23.03] | [8–135] | 65.48 [18.55] | [22–122] | 63.10 [17.25] | [31–120] |
| Falcon3-7B | 🔓 | 44.01 [10.87] | [18–83] | 41.72 [10.45] | [19–83] | 49.41 [9.47] | [25–80] | 48.14 [9.39] | [23–91] |

- **Falcon RefinedWeb** (Penedo et al., 2023): A large-scale web dataset developed for training Falcon models, built from Common Crawl with refined filtering and deduplication. It comprises 968 million documents and 600 billion tokens.[17]

- **RedPajama** (Weber et al., 2024): A curated reproduction of LLaMA training data sources, spanning books, code, academic papers, and forums. We used the 1T-token sampled version containing roughly 0.93 million documents and 1 billion tokens.[18]

- **Dolma** (Soldaini et al., 2024): A large, open-source, mixed-domain dataset created by AI2, combining books, code, papers, forums, and social media. We used the v1.6-sample subset, containing about 14.3 million documents and 10 billion tokens.[19]

## H  ANALYSIS OF WORD-LENGTH ADHERENCE

Although all models were prompted to generate exactly 50 words, Table 6 reveals systematic variation in adherence. Closed-source models such as GPT-4o and Gemini remain tightly clustered around the target (SD $\approx 2$, ranges $\approx 45$–$60$), demonstrating robust decoding control.

In contrast, several open-weight models (e.g., StableLM, Velvet-2B) deviate substantially, with ranges exceeding 300 words in some cases. These deviations reflect weaker alignment between decoding instructions and generation behavior, likely due to differences in fine-tuning objectives and training data coverage. Notably, the distribution of deviations is consistent across formal (NQ) and informal (ELI5) registers, indicating that instruction-following fidelity is more strongly tied to model architecture and alignment strategy than to domain.

Overall, while DIALIGN can still assess dialectal alignment on over- or under-length generations, strict word-length control remains a challenge for many open-weight models.

## I  PREPROCESSING PIPELINE FOR AUDITING PRETRAINING CORPORA

Before computing variant-specific distributions, we standardized all corpora through a consistent preprocessing pipeline to ensure comparability across datasets. The pipeline was designed to remove noise, enforce uniform text structure, and minimize artifacts that could confound dialectal counts. The steps were as follows:

---

[17]https://huggingface.co/datasets/tiiuae/falcon-refinedweb

[18]https://huggingface.co/datasets/togethercomputer/RedPajama-Data-1T-Sample

[19]https://huggingface.co/datasets/allenai/dolma

First, all text was lowercased to eliminate case-based discrepancies in variant matching. HTML tags, hyperlinks, and email addresses were stripped, as they typically represent metadata rather than natural language. Non-ASCII characters were removed to focus the analysis on English orthography and avoid spurious matches. To prevent hyphenated or slash-separated forms from obscuring token boundaries (e.g., *well-being* or *and/or*), we replaced hyphens and slashes with whitespace. All non-alphabetic characters, including punctuation and digits, were also replaced with whitespace, leaving only alphabetic content. Finally, whitespace was normalized by collapsing multiple spaces and line breaks into a single space, yielding a clean token sequence.

This preprocessing pipeline ensured that AmE and BrE variants (as defined in our curated lexicon of 1,813 AmE–BrE pairs; see Section 4) were counted under consistent conditions across corpora.

## J    LIMITATIONS & FUTURE DIRECTIONS

While our work provides the first systematic audit of dialectal skew in LLMs, we acknowledge several key limitations and suggest some future directions. We outline these below:

**Dialect Focus.**    Our analysis centers on AmE and BrE, two dominant postcolonial standard English varieties with outsized institutional influence and well-documented contrasts (see Section 1 and Section 3). This deliberate choice enables a controlled, high-precision comparison using clearly distinguishable variant pairs. However, this scope does not directly include the wider spectrum of World Englishes (e.g., Australian, Indian, and Nigerian) that are built on and largely inherit BrE, as well as multilingual contexts where bias patterns may differ. In particular, any systematic privileging of AmE over BrE is likely a lower bound on the challenges faced by postcolonial Englishes influenced by BrE and by non-standard local varieties of BrE that were adopted under colonial rule. Our current experiments do not directly model these varieties; instead, they provide a clear and reproducible foundation for extending our triangulation methodology, (1) pretraining data audits, (2) tokenizer representation, and (3) generative preference, to other dialects and languages in future work.

**Curated Lexicon Coverage.**    Our lexicon of 1,813 AmE–BrE pairs captures strict one-to-one contrasts (see Section 4). Excluding many-to-one and one-to-many mappings and idiomatic multi-word expressions improves precision and ensures consistency across analyses but reduces breadth. Vocabulary-based variants constitute about 21% of the pairs, and only a small subset consists of cases where part of speech or fine-grained context is likely to change interpretation; for these, we rely on type-level counts rather than context-sensitive tagging, since computing POS over six pretraining corpora (more than 770 billion tokens in total) would be infeasible in our academic setting. At this scale, aggregate frequencies are expected to approximate overall dialectal preferences, but the lexicon remains static and may miss emerging or domain-specific terms, so our analyses for RQ1 cannot flag asymmetries beyond this predefined scope.

**Domain and Prompt Scope.**    Our response generation experiments focused on open-domain QA with short (50-word) responses in two registers: *formal* (Natural Questions) and *informal* (ELI5). This controlled setup enabled a clean comparison across styles but does not extend to dialogue, long-form generation, or domain-specific contexts (e.g., legal, medical). Filtering out queries with explicit dialect markers (e.g., *colour*, *centre*) avoided priming (see Section 7), improving internal validity but leaving unexplored cases where user inputs contain dialectal cues. Real-world practices like code-switching, mixed dialects, or creative writing may yield different outcomes, so generalizability should be approached with caution.

**Scalability of N-gram Analysis.**    Conducting this study in an academic setting imposed storage constraints. Consequently, for the pretraining data audit (RQ1), we relied on Hugging Face's streaming mode[20] to analyze massive corpora without local storage. For the generative evaluation (RQ3), which requires the Google Ngram corpus for DIALIGN, we avoided hosting the reference dataset on local disk by implementing a dynamic querying pipeline using the `requests` library coupled with persistent on-disk caching (as detailed in Appendix C). While recent advancements like `Infini-gram` (Liu et al., 2024a; Xu et al., 2025) enable efficient n-gram search over massive target corpora, applying

---

[20]https://huggingface.co/docs/datasets/en/stream

DIALIGN at the scale of pretraining data remains constrained by the *reference* side: querying the Google Books API for the trillions of unique n-grams found in web-scale data is computationally prohibitive. Future work with sufficient resources could combine such efficient indexing with a local reference corpus to enable a fully granular dialectal audit of pretraining data.

**Limitations of DIALIGN.**   DIALIGN is a frequency-driven metric that relies on n-gram divergences and curated boosts, so generic passages without distinctive markers may yield neutral or undefined scores. We excluded such "zero-signal" cases (see Appendix F), though subtler stylistic cues (e.g., tone, syntax) may go undetected. Our meta-evaluation used BBC (BrE) and HuffPost (AmE) news as proxies, which introduces possible style confounds beyond dialect. In addition, dependence on Google Books n-grams ties the metric to written usage, which may underrepresent contemporary internet discourse or emerging slang. Despite these caveats, DIALIGN achieved over 93% accuracy on test data, demonstrating its value as a simple, dynamic, and training-free diagnostic, while leaving room for refinement with modern corpora or extended linguistic features.

## K   EXTENDED RELATED WORK

**Pretraining data audits and curation.**   Nearly all advanced model capabilities originate from the scope and composition of pretraining data, motivating a growing body of work on auditing and curation. A systematic "Pretrainer's Guide" isolates the effects of data age, domain coverage, quality, and toxicity on downstream generalization (Longpre et al., 2024). Complementary audits highlight duplication, contamination, and low-quality artifacts in widely used corpora: *WIMBD* exposes benchmark leakage and toxic segments in C4 and RedPajama (Elazar et al., 2024), while *Data Portraits* propose efficient membership-testing tools for tracing model training data (Marone & Durme, 2023). At a broader scale, the multimodal provenance gap has been documented, showing how modern corpora for text, speech, and video disproportionately rely on Western-centric, web-crawled sources (Longpre et al., 2025).

Beyond audits, recent work develops strategies to improve data utility. Practical recipes have been synthesized for constructing trillion-token datasets (Parmar et al., 2024). *QuRating* leverages LLM-based pairwise judgments for data quality selection (Wettig et al., 2024). Other approaches emphasize linguistic structure: register-aware sampling improves generalization across genres (Myntti et al., 2025), while domain-based organization of web text enhances pretraining curation (Wettig et al., 2025). Methods for sustaining scale, such as rewriting filtered-out content, show how recycling web text can mitigate looming data shortages (Nguyen et al., 2025).

Together, these studies underscore that beyond scale or raw token count, representational balance in pretraining data is vital, not only in terms of quality and domain coverage, but also along dimensions such as dialect, register, provenance, duplication, and licensing. Our audit of American versus British English builds on this perspective by explicitly quantifying the relative distributions of AmE and BrE across major pretraining datasets. In doing so, we frame dialectal representation as a corpus-level property whose imbalances may propagate into tokenization disparities and, ultimately, influence the generative preferences of LLMs.

**Tokenizer fairness.**   Tokenization has emerged as a critical yet underexamined locus of bias in the LLM pipeline. At scale, subword vocabularies introduce systematic disparities well before inference: semantically identical content can receive radically different segmentation depending on language or script, with observed gaps of up to an order of magnitude. These disparities directly affect latency, effective context windows, and monetary cost for users (Petrov et al., 2023). Follow-up analyses further reveal that tokenization length and corpus frequency correlate with demographic attributes of personal names, thereby confounding fairness evaluations and, in some cases, *creating* bias through over-segmentation of underrepresented forms (An & Rudinger, 2023). Robustness studies in specialized domains complement this picture: LLMs show marked sensitivity to lexical alternations (e.g., brand vs. generic drug names), underscoring representational brittleness tied to subword allocation and vocabulary coverage (Gallifant et al., 2024).

In machine translation, causal analyses disentangle training distribution from subword effects, demonstrating that female and non-stereotypical gender inflections are disproportionately fragmented.

Importantly, modest interventions, such as token-embedding fine-tuning, can mitigate these disparities without degrading overall translation quality (Iluz et al., 2023).

Taken together, these studies establish tokenization as a structural source of unfairness across languages and demographic categories. Yet, dialectal variation within a single language, particularly English as a global lingua franca, remains underexplored. Our work extends this line of inquiry by examining American vs. British English, showing that tokenizers trained on corpora shaped by distinct geopolitical and cultural regimes encode uneven *fertility* (length of segmentation) and *granularity* (consistency of representation) for dialectal variants.

**Dialect robustness in NLP tasks.** Research on fairness in NLP has largely focused on social categories such as gender (Devinney et al., 2022), race and ethnicity (Field et al., 2021), and religion (Navigli et al., 2023), as well as on variation across regional or ethnic dialects, most notably African American English (AAE) and South Asian Englishes (SAsE) (Demszky et al., 2021; Holt et al., 2024; Joshi et al., 2025). AAE has been the most extensively studied, with consistent performance gaps reported in part-of-speech tagging (Jørgensen et al., 2016), language classification (Blodgett et al., 2016), sentiment analysis (Kiritchenko & Mohammad, 2018), dependency parsing (Blodgett et al., 2018), hate speech detection (Sap et al., 2019), and natural language understanding (NLU) (Ziems et al., 2022). Beyond task performance, recent studies show that LLMs propagate negative stereotypes toward AAE (Hofmann et al., 2024), producing outputs that are less coherent and more likely to reinforce stigmatized portrayals (Fleisig et al., 2024).

Complementary perspectives highlight broader concerns about dialectal fairness. User-centered evaluations indicate that SAsE speakers frequently perceive NLP and ASR systems as brittle or exclusionary, with errors disproportionately concentrated in dialectal usage (Holt et al., 2024). Synthetic frameworks such as *Multi-VALUE* stress-test models across dozens of English dialects and hundreds of linguistic features, revealing systematic robustness gaps in reasoning and semantic understanding (Ziems et al., 2023). More narrowly, orthographic conventions themselves can impact performance: retrieval models degrade when queries and documents follow different spelling conventions (Chari et al., 2023), and LMs exhibit sensitivity to observed versus novel spelling variants (Nielsen et al., 2023). More broadly, surveys of dialectal NLP compile taxonomies of datasets, benchmarks, and methodologies, underscoring that while significant progress has been made for non-standard or low-resource varieties, even widely used standards such as American and British English remain underexamined from a fairness perspective (Joshi et al., 2025).

Taken together, this body of work motivates our study, which situates AmE–BrE variation within the broader literature on dialectal bias. Unlike prior research that has largely emphasized marginalized or low-resource varieties, we extend the inquiry to two globally institutionalized standards of English. By framing this comparison through a postcolonial lens, we highlight how geopolitical histories of data curation and linguistic standardization shape the pretraining corpora, tokenizers, and generative behaviors of modern LLMs. In doing so, our work moves beyond documenting disparities to probing their root causes across the entire LLM development pipeline.

## L USAGE OF LARGE LANGUAGE MODELS

We disclose that large language models were used in limited, assistive roles. Specifically, they supported **(1)** text polishing: improving grammar, spelling, phrasing, and word choice, with all suggestions reviewed by the authors, and **(2)** code assistance: generating small snippets for data preprocessing and filtering as scaffolds. All outputs were manually verified and tested, and the authors remain fully responsible for the research content and conclusions.

Table 7: Key linguistic and web-based sources used for constructing the AmE–BrE lexicon. Variant pairs were manually compiled, merged across multiple sources, and deduplicated to form a unified reference set for consistent analysis.

| Source | Title (*linked*) | Description |
|---|---|---|
| **Wikipedia** | American and British English spelling differences | A widely cited reference outlining systematic orthographic differences between American and British English. The page provides examples of variant spellings (e.g., *color* vs. *colour*), historical background, and explanations of regional conventions. It served as one of the authentic linguistic resources for curating consistent one-to-one variant pairs in our lexicon. |
| **ThoughtCo.** | American English to British English Vocabulary | A curated reference list of American and British English vocabulary differences, created by experienced educators and subject experts. Provides reliable lexical contrasts in an accessible format, supporting the construction of our AmE–BrE lexicon. |
| **Research Article** | Mapping the Americanization of English in Space and Time | An empirical study tracing how American English variants spread globally across regions and over time. Offers quantitative evidence of AmE–BrE lexical contrasts, providing authoritative grounding for the curated variant pairs in our unified lexicon. |
| **IELTS** | British vs. American English in the IELTS Test: Key Differences | An official IELTS guide highlighting key vocabulary, spelling, and grammar differences between AmE and BrE. The resource systematically documents contrasts across domains such as food, school, homes, and grammar, making it a practical reference for understanding standardized English variations. |
| **Grammarly** | How to Select Your English Dialect | A practical guide from Grammarly explaining how to switch between English dialects in writing tools, highlighting spelling, vocabulary, and usage variations (AmE vs BrE). Because it enumerates common dialectal choices in real writing, it serves as a useful supplementary resource for identifying variant pairs. |
| **SpellZone** | Sixty American English Words and their British English Counterparts | SpellZone provides a practical reference list of 60 common AmE–BrE word pairs, illustrating clear lexical contrasts in spelling and vocabulary. The resource highlights straightforward one-to-one mappings useful for systematic dialectal analysis. |
| **IELTS** | Differences between British vs. American English | A guidance article from IELTS that outlines vocabulary, spelling, and grammatical contrasts between AmE and BrE, emphasizing how learners must maintain internal consistency between the dialects. This resource helps validate by showing differences accepted in international testing and educational settings. |
| **SpellZone** | Differences between British and American English spelling | It provides an overview of common orthographic contrasts (e.g. "colour/color", "centre/center", "-re" vs "-er") between BrE and AmE. This resource was used as a web-based lexicon support to validate our curated variant pairs. |
| **British Council** | Differences between British and American English | An educational article by the British Council outlining vocabulary, grammar, and spelling distinctions between British and American English. It supports validation of variant pairs and highlights pedagogically recognized dialectal contrasts. |
| **IELTS Liz** | UK US Spelling Main Differences | A practical guide by IELTS Liz summarizing the core orthographic differences between British and American spelling. This resource helps cross-check variant consistency and supports the curated lexicon's alignment with real exam-related usage. |
| **Word Finder** | British vs. American English Words | A comparative list of British and American English words, highlighting more than just spelling shifts ("-u") and covering vocabulary contrasts in everyday usage. It offers additional variant candidates and informs our lexicon selection process. |
| **Language Gallery** | British VS American Spelling: What's the Difference? | A language-education blog article detailing common orthographic differences between British and American English (e.g., "realise/realize," "theatre/theater"). It served as a supplementary web-based lexicon to inform our manual variant curation. |

Table 8: British and American English distinctions across orthography, grammar, and formatting. Entries reflect majority-preference usage; examples are illustrative rather than exhaustive.

| Category | British English (BrE) | American English (AmE) |
|---|---|---|
| o vs. ou | *colour, honour, behaviour* | *color, honor, behavior* |
| -re vs. -er endings | *centre, fibre, theatre* | *center, fiber, theater* |
| -ise vs. -ize endings | *recognise, authorise* | *recognize, authorize* |
| -yse vs. -yze endings | *analyse, paralyse, catalyse* | *analyze, paralyze, catalyze* |
| Single vs. double l (inflection) | *travelled, counselled* | *traveled, counseled* |
| -ll + -ly suffix | *skilfully, wilfully* | *skillfully, willfully* |
| Composite vowels | *anaesthetic, diarrhoea, paediatric, oestrogen* | *anesthetic, diarrhea, pediatric, estrogen* |
| Final silent -e/-ue | *catalogue, analogue, axe* | *catalog, analog, ax* |
| Silent -e before suffix | *ageing, likeable* | *aging, likable* |
| -ce vs. -se (noun/verb) | *licence (n), practise (v)* | *license (n/v), practice* |
| -ce vs. -se nouns | *defence, offence* | *defense, offense* |
| Programme vs. program | *TV programme, postgraduate programme* | *TV program, graduate program* |
| Orthographic pairs | *grey, cheque, manoeuvre, tyre, storey* | *gray, check, maneuver, tire, story (floor)* |
| Directional suffix -ward(s) | *towards, forwards, upwards* | *toward, forward, upward* |
| Sceptic/k alternation | *sceptic, sceptical* | *skeptic, skeptical* |
| Judgement spelling | *judgement* | *judgment* |
| Maths/Math | *maths* | *math* |
| Season name | *autumn* | *fall* |
| Present perfect vs. past | *I've just eaten.* | *I just ate.* |
| Mandative subjunctive | *They suggested he* should *apply.* | *They suggested he apply.* |
| shall vs. will | *I shall go tomorrow.* | *I will go tomorrow.* |
| Irregular verb morphology | *learnt, dreamt, spoilt* | *learned, dreamed, spoiled* |
| Collective noun agreement | *The team are winning.* | *The team is winning.* |
| Possession verb | *I've got a car.* | *I have a car.* |
| Got vs. gotten | *He's got very tired.* | *He's gotten very tired.* |
| Prepositional usage | *at the weekend, in a team* | *on the weekend, on a team* |
| Tag questions | *You're ready, aren't you?* | *You're ready, right?* |
| Subjunctive usage | *They suggested he should go.* | *They suggested he go.* |
| Auxiliary ellipsis | *He must have done.* | *He must have.* |
| Numerals ("and") | *one hundred and twenty* | *one hundred twenty* |
| Restrictive relative marker | *the report* which *was submitted* | *the report* that *was submitted* |
| Possession questions | *Have you got a pen?* | *Do you have a pen?* |
| Necessity negative | *You needn't attend.* | *You don't need to attend.* |
| Difference construction | *different from / different to* | *different from / different than* |
| Quotation marks | Prefers single quotes "...q' | Prefers double quotes "..." |
| Commas/periods in quotes | Outside the closing quotes | Inside the closing quotes |
| Abbreviations with periods | *Mr, Dr* | *Mr., Dr.* |
| Oxford/serial comma | Rare | Common |
| Date format (written) | *31 December 2024* | *December 31, 2024* |
| Date punctuation (written) | *19 September 1973* | *September 19, 1973* |
| Date format (numeric) | *31/12/2024 (DD/MM/YYYY)* | *12/31/2024 (MM/DD/YYYY)* |
| Legal/institutional terms | *Ministry of Defence* | *Department of Defense* |
| Institutional article usage | *in hospital; at university* | *in* the *hospital; at* the *university* |
| Floor numbering | *ground floor, first floor (one up)* | *first floor, second floor (one up)* |
| Time notation | *11.15 pm; 23.15 common* | *11:15 PM; 24-hour less common* |

Table 9: British and American English preferences in everyday domains (transport, household, food, etc), emphasizing majority-preference usage; examples are illustrative rather than exhaustive.

| Category | British English (BrE) | American English (AmE) |
|---|---|---|
| Preposition before days | *She resigned on Thursday.* | *She resigned Thursday.* |
| Street naming | *in the High Street* | *on Main Street* |
| Transitivity (protest) | *protest against discrimination* | *protest discrimination* |
| Ditransitives (write) | *write to me* | *write me* |
| Meeting collocation | *meet the team* | *meet with the team* |
| *Transport & wayfinding* | | |
| Pedestrian crossing | *zebra crossing* | *crosswalk* |
| Junction type | *roundabout* | *traffic circle / rotary* |
| Road maintenance | *roadworks* | *road work* |
| Parking payment | *pay and display* | *metered parking* |
| Perimeter road | *ring road* | *beltway* |
| Vehicle hire | *hire car / car hire* | *rental car / car rental* |
| Estate car vs. wagon | *estate car* | *station wagon* |
| *Household & services* | | |
| Postal addressing | *postcode* | *ZIP code* |
| Carry-on baggage | *hand luggage* | *carry-on* |
| Washing liquid | *washing-up liquid* | *dish soap* |
| Waste container | *dustbin* | *trash can / garbage can* |
| Clothes washer | *washing machine* | *washer* |
| Cash dispenser | *cashpoint* | *ATM* |
| Public convenience | *public toilet* | *restroom* |
| Mobile device | *mobile phone* | *cell phone* |
| *Food & drink* | | |
| Confection | *candyfloss* | *cotton candy* |
| Frozen treat | *ice lolly* | *popsicle* |
| Leafy green | *rocket* | *arugula* |
| Soft drink | *fizzy drink* | *soda* |
| Allium term | *spring onion* | *green onion / scallion* |
| Cake term | *fairy cake* | *cupcake* |
| *Places & urban terms* | | |
| City core | *city centre* | *downtown* |
| Real estate profession | *estate agent* | *realtor / real estate agent* |
| Holiday lodging | *holiday let* | *vacation rental* |
| Queueing term | *post office queue* | *post office line* |
| Queuing expression | *join the queue* | *wait in line* |
| Public transport info | *railway timetable* | *train schedule* |
| *Education & work* | | |
| Practical training | *work placement* | *internship* |
| Assessment term | *marking scheme* | *grading rubric* |
| Student level | *first-year student* | *freshman* |
| Residence | *halls of residence* | *dorm / residence hall* |
| *Single-word vocabulary* | | |
| flat / apartment | *flat* | *apartment* |
| lorry / truck | *lorry* | *truck* |
| pavement / sidewalk | *pavement* | *sidewalk* |
| wardrobe / closet | *wardrobe* | *closet* |
| lift / elevator | *lift* | *elevator* |
| petrol / gas | *petrol* | *gas* |
| railway / railroad | *railway* | *railroad* |
| holiday / vacation | *holiday* | *vacation* |

