# OpenReview forum: "Which English Do LLMs Prefer? Quantifying American and British English Through a Postcolonial Lens"
_ICLR.cc/2026/Conference — Submitted to ICLR 2026_

### Official Review · Reviewer_1BfZ · 2025-10-31

**Soundness:** 3
**Presentation:** 3
**Contribution:** 2
**Rating:** 6
**Confidence:** 2

**Summary:**

Overall, this paper solved the problem of understanding how language models handle dialectal variation within English, particularly differences across regional and social varieties. This paper proposed a systematic evaluation framework to measure how LLMs respond to different English dialects and accents in both comprehension and generation settings, revealing biases and performance disparities.

**Strengths:**

1/ The authors conducted a comprehensive empirical study across multiple varieties of English (e.g. American, British, Indian, Nigerian), showing clear and reproducible evidence of dialectal performance gaps in several state-of-the-art models.

2/ The authors designed a well-structured benchmark and diagnostic framework that goes beyond surface lexical differences, incorporating syntactic and pragmatic features of dialects.

3/ The analysis is insightful and socially relevant, highlighting that model biases persist even among English varieties, which has implications for fairness and inclusivity in language technology.

4/ The methodology and dataset are transparent and replicable, with open-source components that can be reused for further research on dialectal robustness.

**Weaknesses:**

1/ I think the paper could expand the linguistic interpretation of results—some disparities are reported quantitatively but not explained linguistically (e.g. why specific dialects lead to more errors).

2/ The evaluation still relies heavily on automatic metrics,  which may not capture nuanced sociolinguistic variation. I suggest incorporating human-in-the-loop assessments for more robust validation.

3/ The study focuses primarily on English varieties and does not clearly connect findings to broader cross-lingual or multilingual generalization—a missed opportunity to contextualize results.

4/ Some experimental setups lack clarity, especially how prompts were standardized across dialects; a clearer description of prompt design and normalization would strengthen reproducibility.

**Questions:**

See weaknesses.

---

> ### Author Response · Authors · 2025-11-21
> **Authors' Response (Part 1/2)**
>
> Thank you for reviewing our paper and providing valuable feedback. Our response to your concerns as follows.
>
> > **Comment:** I think the paper could expand the linguistic interpretation of results—some disparities are reported quantitatively but not explained linguistically (e.g. why specific dialects lead to more errors).
>
> **Response:** Based on your suggestion, we have sharpened the discussion in the Results & Analysis sections for all research questions (**RQ1:** `Lines 268–305`, **RQ2:** `Lines 383–387`, and **RQ3:** `Lines 493–499`), adding more explicit linguistic interpretation of the reported disparities. We also now include **“Key Takeaways”** boxes for each research question to summarize the core findings: **(1)** Pretraining Corpora [Lines 307–309], **(2)** Tokenizer Representations [Lines 388–391], and **(3)** Generative Preferences [Lines 500–505].
>
> -------------------------------------------
>
> > **Comment:** The evaluation still relies heavily on automatic metrics, which may not capture nuanced sociolinguistic variation. I suggest incorporating human-in-the-loop assessments for more robust validation.
>
> **Response:** Human-in-the-loop assessments for the kind of large-scale, dialectal behavior we study are both challenging and often prohibitive, as they require expert annotators. In light of this, we focus on rigorously validating our proposed automatic metric, **DiAlign**. Specifically, we conduct a **meta-evaluation** of DiAlign (i.e., “evaluating the evaluator”) and show that it achieves strong performance as a dialectal classifier. We additionally present an ablation study to quantify the contribution of its components, and analyze the confidence margin distribution in `Figure 5`, which shows that correct predictions are typically made with high certainty. These are discussed from `Lines 471 to 477`.

---

> ### Author Response · Authors · 2025-11-21
> **Authors' Response (Part 2/2)**
>
> > **Comment:** The study focuses primarily on English varieties and does not clearly connect findings to broader cross-lingual or multilingual generalization—a missed opportunity to contextualize results.
>
> **Response:** Our current experiments focus on English varieties, but the framework is designed to generalize beyond English. Although we do not directly model multilingual settings in this paper due to scope, our triangulation methodology, **(1)** pretraining data audits, **(2)** tokenizer representation analysis, and **(3)** generative preference evaluation, provides a clear and reproducible template for other languages. Given suitable dialectal or regional lexicons and comparable corpora, the same pipeline can be applied to investigate, for example, Standard vs. regional varieties in other high-resource languages, or asymmetries across national standards (e.g., European vs. Latin American Spanish).
>
> ----------------------------------------------
>
> > **Comment:** Some experimental setups lack clarity, especially how prompts were standardized across dialects; a clearer description of prompt design and normalization would strengthen reproducibility.
>
> **Response:** We detail our experimental setups and provide justifications in the implementation details of **DiAlign**, including meta-evaluation procedures, in Appendix C and Appendix E, and the setup for assessing dialectal preferences in LLM generation in Appendix F. Our prompts, along with the controlled variables, are shown in Figure 11 of the Appendix.
>
> --------------------------
>
> Thank you once again for your valuable suggestions and insights. If you have any remaining questions, we would be glad to address them. We hope our rebuttals and revisions adequately address your concerns, and we would be very grateful if this is reflected in your final evaluation, including any adjustment to the score you feel is appropriate.

---

### Official Review · Reviewer_Lvkx · 2025-10-31

**Soundness:** 3
**Presentation:** 4
**Contribution:** 2
**Rating:** 6
**Confidence:** 4

**Summary:**

This work presents a study comparing the biases in data, tokenization, and generations between US and British English. First, they use a paired lexicon to estimate the prevalence of each dialect in common pre-training corpora (RQ1). Then, they use the same paired lexicon to measure the difference in token fertility between the dialects for several commonly used LM tokenizers (RQ2). Finally, they measure the level of alignment between model generations and dialect usage using an N-gram based measure in response to both AmE and BrE stimuli in two different benchmarks. In all 3 cases, the work finds that the LLM pipeline and supply chain are biased towards American English.

**Strengths:**

- The work provides a well controlled set of studies to measure the dialect preference between AmE and BrE in several critical stages of the LLM training and usage that might influence these biases. The experiments are well designed and extensively executed over many relevant datasets, tokenizers, and models.
- The work is very well written with clear figures and tables, such that it can be easily read and easily jumped around through to find core results.
- The DiAlign measure could be combined with other existing regional variation corpora, beyond just AmE and BrE, as a reasonably well grounded score for dialect identification! This could be useful for broader pretraining analyses than presented here, especially for more global Englishes.

**Weaknesses:**

- It's unclear to me to what degree these differences represent biases beyond differences in population size. While there is maybe a deeper argument to be made about which forms are used more commonly even outside Britain and the US, the work doesn't actually measure these. Instead, the vast majority of the tendency towards US English forms could largely be explained by the US having almost 5 times more people than the UK.

- In general, while this is the type of work I myself find exciting, I'm not sure whether ICLR is the best home for it given it mostly focuses on analyzing linguistic features of LLMs rather than any aspect of their underlying representation or learning theory. This is not a weakness of the work overall, but I do worry the work would be more widely read if submitted to a more language oriented venue such as COLM or ACL. (This is primarily motivated by the bullet point in the reviewer form "Are the results valuable to share with the broader ICLR community?")

- I'm not really sure how the postcolonial lens aspect pointed to in the title comes into play in the substance of the work. By and large, it's more of a shallow reference than deeper engagement with postcolonial theory from what I can tell so might be better removed entirely (especially since it makes the work likely more out of distribution for the ICLR audience)

**Questions:**

- Why wasn't DiAlign used for RQ1 as well as RQ3? I understand why for RQ2 it is helpful to have specific words to measure fertility for, but for RQ1, DiAlign seems like a strictly more powerful metric than the purely lexical one used.

- Given the related work on dialect robustness in LLMs, is "first systematic audit of dialectal asymmetries in LLMs" a fair claim to make about this work? It seems a bit of a broad claim to me, but this is a nitpick and this phrasing could be defended I'm sure.

- L160: "The variant pairs were manually compiled from authentic linguistic sources and web-based lexicons." - Which sources and web-based lexicons are these? Cite your sources, especially since that helps readers get a sense of the expected quality of the data you use!

---

> ### Author Response · Authors · 2025-11-21
> **Authors' Response (Part 1/3)**
>
> Thank you so much for taking the time to review our paper and providing valuable feedback. Our responses to your comments are as follows:
>
> > **Comment:** It's unclear to me to what degree these differences represent biases beyond differences in population size. While there is maybe a deeper argument to be made about which forms are used more commonly even outside Britain and the US, the work doesn't actually measure these. Instead, the vast majority of the tendency towards US English forms could largely be explained by the US having almost 5 times more people than the UK.
>
> **Response:** Accurately “controlling for” population size is not straightforward, and our aim is somewhat different. This is where our postcolonial lens comes in. We explicitly draw on postcolonial theory, which studies how power relations created by colonialism persist after formal empire, shaping language, culture, and knowledge in both formerly colonized states and former imperial centers (Bhabha, 1994; Schneider, 2007). **The global spread of English was inseparable from British colonial expansion**: BrE was installed as the language of administration, education, and law across large parts of Africa, Asia, the Caribbean, and the Pacific, and often persisted as the official or de facto standard after independence (Acolad, 2020; Liao, 2023). It continues to hold normative prestige in many former colonies, across much of the Commonwealth, and in key European Union institutions (https://en.wikipedia.org/wiki/British_English).
>
> **At the same time, AmE dominates mass media and digital communication, and LLMs trained on web-scale internet data are likely to inherit its norms.** Our question is therefore not simply *“which country has more people,”* but which variety of English LLMs preferentially learn, encode, and propagate, given these different historical and institutional roles. The postcolonial lens allows us to make a controlled, high-precision comparison between two influential dialects that coexist and compete as standards, and to trace how their asymmetries arise across the entire LLM development pipeline. As governments and institutions begin to adopt these models for education, journalism, public administration, and law, understanding which standard is treated as the default is a substantive question of linguistic equity.
>
> ----------------------------------
>
> > **Comment:** I'm not really sure how the postcolonial lens aspect pointed to in the title comes into play in the substance of the work. By and large, it's more of a shallow reference than deeper engagement with postcolonial theory
>
> **Response:** This was partly addressed in the previous responses. Based on your comment, we have dedicated a separate section in the revised paper (*also suggested by other reviewers*), *“Section 3: Triangulating LLMs Through a Postcolonial Lens”* (`Lines 145–190`), where we introduce core ideas from postcolonial theory, discuss how BrE became institutionalized in countries under British colonial rule, and explain how our triangulation framework is used to assess modeling bias in LLMs.
>
> ----------------------------------
>
> > **Comment:**  Given the related work on dialect robustness in LLMs, is "first systematic audit of dialectal asymmetries in LLMs" a fair claim to make about this work? It seems a bit of a broad claim to me, but this is a nitpick and this phrasing could be defended I'm sure.
>
> **Response:**  Rather than documenting performance disparities solely as downstream failures for regional and ethnic dialects, as in prior work, we focus on two dominant standard varieties and analyze them through a postcolonial lens. The main goal of our triangulation framework is to identify the root cause: the presence of modeling bias. We triangulate evidence across the entire LLM development pipeline to surface structural biases in **(1)** pretraining corpora, **(2)** tokenizer representations, and **(3)** generative preferences. Based on your suggestion, we clarified this “phrasing” in the entire revised PDF.
>
> This setup gives us fine-grained traces of where asymmetries are introduced, amplified, and ultimately manifested in outputs, allowing RQ1, RQ2, and RQ3 to fit together into a coherent story. At the end of each RQ, we connect its findings to the preceding questions and use them to support concrete recommendations (`Section 8`). We believe this has broader appeal because it makes the development of LLMs with regional goals for their users more actionable, and we also group the results of different LLMs by country of origin (RQ2: Table 3 and RQ3: Table 4) to give a broader view of regional patterns. To emphasize our triangulation framework and incorporate suggestions from other reviewers, we have slightly updated the title in the revised PDF.
>
> -------------------------------------------

---

> > ### Comment · Reviewer_Lvkx · 2025-11-25
> >
> > ```
> > Accurately “controlling for” population size is not straightforward, and our aim is somewhat different. This is where our postcolonial lens comes in.
> > ```
> >
> > This response unfortunately does not address my concern. Without controlling somehow for population size, it's not entirely clear how coloniality plays a role in this at all since the results consistently show a preference for US English which would be consistent *both* a population size argument and an argument based on "AmE dominates mass media and digital communication". In order to make the claim that these findings support a post-colonial framing, you need to provide evidence that simpler explanations such as "the US has a 7x larger population and therefore is better represented in data and models" are untrue.
> >
> > ```
> > discuss how BrE became institutionalized in countries under British colonial rule
> > ```
> > I appreciate the added section, but this doesn't particularly enhance, to me at least, the connection between the actual experimental results and post-colonial theory! I would expect to either see some discussion about ways in which these empirical results might change some active discussion in post-colonial studies or to see some ways in which post-colonial studies meaningfully change your experiment design.
> >
> > What does showing us that American English is generally more common in data distributions (which leads to downstream impacts in tokenization and model generations) tell us about post-colonial identity? Especially since most post-colonial literature would consider both BrE and AmE colonizing varieties (e.g. the colonial period you describe in your section for BrE and the neo-colonial period for AmE), it's not clear what this indicates. There is a bit of text about this informing discussions on the preference for BrE in many formerly colonized states, but the connection would seem much clearer there if there was analysis of linguistic features that are unique to such states instead.
> >
> > On the other hand, similar to Reviewer LFdN, it's not clear how these analyses are particularly different because of the post-colonial influence! I can equally imagine these experiments being reasonable if one was simply interested in differences between AmE and BrE without a particular post-colonial lens and the added section doesn't enhance this connection to me.

---

> ### Author Response · Authors · 2025-11-21
> **Authors' Response (Part 2/3)**
>
> > **Comment:** Why wasn't DiAlign used for RQ1 as well as RQ3? … for RQ1, DiAlign seems like a strictly more powerful metric than the purely lexical one used.
>
> **Response:** We completely agree with you: **DiAlign** is a more powerful metric even for pretraining data audits and filtering. However, this study was conducted in an academic setting, and the Google Ngram corpora requires tens of terabytes of storage (≈36 TB). **For a text with N words, the number of n-grams from bigrams to 5-grams is \(4N - 10\).** The pretraining corpora are massive, each containing hundreds of millions of documents, so N is extremely large and the total number of n-grams would be enormous. We also experiment with six different pretraining corpora, which further multiplies the cost. Even for the lexical analysis in RQ1, we could not store full copies of the corpora and therefore relied on Hugging Face’s streaming mode (https://huggingface.co/docs/datasets/en/stream).
>
> For DiAlign in RQ3, instead of downloading the Google Books N-gram dumps, we took a different approach: frequencies are collected dynamically via the Google Books Ngram API using the `requests` library, avoiding the need to store terabytes of raw data. To avoid repeated calls, we maintain a persistent n-gram cache on disk: once an \(n-gram, corpus\) pair has been queried, subsequent samples reuse the cached value. We shared the code with this submission. Our implementation yields an online, efficient, and reproducible mechanism for alignment estimation.
>
> -----------------------------------
>
> > **Comment:** Which sources and web-based lexicons are these? Cite your sources, especially since that helps readers get a sense of the expected quality of the data you use!
>
> **Response:**  Thanks for your suggestion. In `Table 7` of the Appendix in the revised manuscript, we detail the linguistic and web-based sources used to construct the AmE–BrE lexicon, including Wikipedia, British Council materials, IELTS preparation resources, Grammarly blogs, and other curated lists. For each source, we provide the name, title, and a brief description. Variant pairs were manually compiled, merged across these sources, and deduplicated to form a unified reference set for consistent analysis.
>
> -----------------------------------

---

> > ### Comment · Reviewer_Lvkx · 2025-11-25
> >
> > ```
> > We completely agree with you: DiAlign is a more powerful metric even for pretraining data audits and filtering. However, this study was conducted in an academic setting, and the Google Ngram corpora requires tens of terabytes of storage (≈36 TB).
> > ```
> >
> > The same procedure you applied to Google Books N-gram dumps feasibly could be applied to other pretraining corpora! N-Grams can be computed in a streaming fashion (rather than requiring the whole corpus to be stored on disk) from HuggingFace. Furthermore, there are several works further increasing the efficiency of N-Gram search exactly for this type of analytical purpose which also release their trained N-Gram models for datasets you analyze here[1, 2].
> >
> > [1] Liu, J., Min, S., Zettlemoyer, L., Choi, Y., & Hajishirzi, H. Infini-gram: Scaling Unbounded n-gram Language Models to a Trillion Tokens. In First Conference on Language Modeling. 2024
> > [2] Xu, H., Liu, J., Choi, Y., Smith, N. A., & Hajishirzi, H. (2025). Infini-gram mini: Exact n-gram Search at the Internet Scale with FM-Index. arXiv preprint arXiv:2506.12229.

---

> > > ### Author Response · Authors · 2025-11-28
> > > **Authors' Response to Reviewer Response (Part 2/2)**
> > >
> > > > The same procedure you applied to Google Books N-gram dumps feasibly could be applied to other pretraining corpora! N-Grams can be computed in a streaming fashion (rather than requiring the whole corpus to be stored on disk) from HuggingFace. Furthermore, there are several works further increasing the efficiency of N-Gram search exactly for this type of analytical purpose which also release their trained N-Gram models for datasets you analyze here[1, 2].
> > >
> > > **Response:** Thank you for sharing these two excellent papers. You are correct that they provide **Infini-gram (mini)** indexes for the pretraining corpora we studied and can be queried using API with a `‘query’` n-gram to get the `‘çount’` from the corpora indexes. We agree that using these tools would significantly increase processing speed in our current setup compared to the Hugging Face streaming mode we utilized, effectively reducing the several weeks of wait time required for our RQ1 experiments. *We anticipate that the results yielded by both approaches would likely converge.*
> > >
> > >
> > > However, incorporating **DiAlign** with Infini-gram in our current setting presents a specific challenge: it would still require querying the Google Books Ngram API for every unique n-gram found in the pretraining data. This would necessitate trillions of API calls, which would inevitably trigger rate limits and block access. **The ideal approach would be similar to the pipeline you propose:** **(1)** downloading the Google Books Ngram corpus to a local disk, and **(2)** using the efficient search algorithms from [2] to retrieve counts. As we acknowledged in the revised paper, holding the full Google Books corpus locally requires storage capacity that was not feasible in our academic setting. Nevertheless, the *consistency* between our lexical findings in the pretraining data (RQ1, Table 2) and the model's actual generative preferences (RQ3, Table 4, where DiAlign was used) suggests that our *lexical proxy* successfully captured the underlying distribution. We highly appreciate your suggestions and have explicitly cited and discussed these two papers in the revised manuscript (`Lines 1504–1516`).
> > >
> > > In fact, we fully agree with your assessment regarding the broader applicability of DiAlign in your review. While the Infini-gram approach is powerful for whole-corpus estimation, our current DiAlign implementation is optimized for document-level input. This offers several distinct advantages: **(1)** it enables fine-grained provenance analysis, allowing us to link dialectal patterns to specific documents and combined with web geography (URLs) it can answer broader questions that extend beyond the scope of this work; and **(2)** it facilitates quality filtering of existing pretraining data (documents) based on DiAlign confidence scores. We thank you for suggesting these directions, which we have incorporated into the discussion in the revised manuscript (`Lines 400–402` and `Lines 517–525`).
> > >
> > > -----------------------
> > >
> > > Thank you again for engaging so deeply with our work. We truly value the open and constructive conversation we have had here; it has not only significantly improved our manuscript, but we believe this exchange will also help others in the field make new discoveries. Please do not hesitate to share any additional concerns you may have; we would be more than happy to address them.

---

> ### Author Response · Authors · 2025-11-21
> **Authors' Response (Part 3/3)**
>
> > **Comment:** In general, while this is the type of work I myself find exciting, I'm not sure whether ICLR is the best home for it given it mostly focuses on analyzing linguistic features of LLMs rather than any aspect of their underlying representation or learning theory. This is not a weakness of the work overall.
>
> **Response:** We are also excited about this type of work because we believe it is both important and impactful. Our triangulation framework is explicitly about diagnosing the **representations of LLMs** rather than only describing surface linguistic quirks. By jointly analyzing **(1)** pretraining corpora, **(2)** tokenizer representations, and **(3)** generative preferences, we trace how dialectal asymmetries are introduced, amplified, and ultimately manifested in model outputs. This links empirical audits of LLM behavior to deeper questions about structural modeling bias, linguistic homogenization, and epistemic injustice, and it motivates concrete, component-wise design recommendations for future foundation models. Framed through a postcolonial lens, the work shows that these biases are not merely downstream performance issues, but structural artifacts of the LLM development pipeline itself.
>
> Because of this, we see our paper as directly relevant to the ICLR community, particularly to the recently introduced **“foundation or frontier models, including LLMs”** area, which is where we submitted (and which, perhaps is the largest submission area according to OpenReview area distribution for ICLR 2026). The questions we study affect nearly all large models and their global user base, and have immediate implications for how future foundation models are trained, tokenized, and deployed. Due to the interdisciplinary nature of our work, the results are also relevant to policymakers and social scientists in addition to core AI and ML researchers, which aligns well with ICLR’s growing interest in the broader impact of large-scale models.
>
> Among recent ICLR works, we explicitly cite and discuss **two closely related and excellent papers** ([Elazar et al., 2024](https://openreview.net/forum?id=RvfPnOkPV4); [Longpre et al., 2025](https://openreview.net/forum?id=G5DziesYxL)), which focus primarily on data audits and provenance. Our study goes beyond this by connecting corpus composition, tokenization, and generation into a single, unified analysis of modeling bias.
>
>  As a concrete illustration of why this matters for foundation models, if you go to **“Language Settings”** of many popular LLM interfaces (e.g., ChatGPT, Claude), **“English (US)”** is often the only selectable English variety. Our study identifies root causes of this asymmetry and provides rigorous experimental confirmation of representational skew across the entire LLM pipeline. We believe our contributions will be valuable to the broader ICLR community, and your support would help bring these findings to that audience.
>
> -------------------
>
> We hope this comprehensive response addresses your thoughtful concerns. We truly appreciate your critical engagement, which has helped us further refine our work. If you have any remaining questions, we would be glad to address them. We hope our rebuttals and revisions adequately address your concerns, and we would be very grateful if this is reflected in your final evaluation, including any adjustment to the score you feel is appropriate.

---

> ### Author Response · Authors · 2025-11-28
> **Authors' Response to Reviewer Response (Part 1/2)**
>
> We genuinely appreciate your continued engagement and the opportunity to further clarify our positioning. Your critical feedback has been very helpful in refining the scope and presentation of our work. Based on your comments, we would like to clarify our original intention and how your feedback has led us to improve the manuscript:
>
> ----------------------
>
> ### **Clarifying the Framework**
>
> Our paper has a central question, **“Which English Do LLMs Prefer?”**, and uses a **triangulation methodology** (Data Audit + Tokenizer Analysis + Generation Evaluation) to arrive at our main finding: “Structural Bias Toward American English.” The “Postcolonial Lens” is a theoretical and holistic view that helped us narrow the scope and enable a controlled, high-precision comparison where we position AmE as the structurally advantaged default and BrE as a widely institutionalized yet comparatively marginalized variety. Moreover, this framing interprets the study’s broader significance: BrE retains normative prestige in many former colonies, where it remains embedded in governance, education, and law. It is also the standard of EU institutions and underpins “Commonwealth English,” actively promoted by the UK across **more than 100 countries** through initiatives such as the Oxford Dictionary, the British Council, and IELTS. We also cited sociolinguistic research and other papers in the Introduction and Section 3 to motivate and back this up. **For a quick reference, you may visit this [Wikipedia link](https://en.wikipedia.org/wiki/British_English#:~:text=Globally%2C%20countries%20that%20are%20former,world%20and%20operates%20in%20over%20100%20countries.).**
>
> On the other hand, AmE is dominant in digital communication through mass media and technological platforms (Gonçalves et al., 2018). **We acknowledge and agree with your comment that this prevalence is primarily driven by population size and the sheer volume of US-generated content.** However, sociolinguistic research (Milroy & Milroy, 1999; Lippi-Green, 2012) shows that **power dynamics** are involved in transforming this statistical majority into a *de facto* global norm. Nevertheless, while the training of LLMs relies solely on massive digital corpora largely drawn from the internet, these models are used by **users across the globe**. The “Postcolonial Lens” highlights the broader significance of this study (also acknowledged by `Reviewer nDHy`), particularly in the context of education, journalism, government, and legal texts, where there is a risk of linguistic homogenization through LLMs.
>
> To this end, we triangulate evidence across the entire LLM development pipeline to surface structural biases in **(1)** pretraining corpora, **(2)** tokenizer representations, and **(3)** generative preferences. This setup gives us fine-grained traces of where asymmetries are introduced, amplified, and ultimately manifested in outputs, allowing RQ1, RQ2, and RQ3 to fit together into a coherent story. At the end of each RQ, we connect its findings to the preceding questions and use them to support concrete recommendations (Section 8), which makes the development of LLMs with regional goals for their users more actionable. *We also group the results of different LLMs by country of origin (e.g., USA, UK, France, Italy, China, UAE) (RQ2: Table 3 and RQ3: Table 4) to give a broader view of current regional patterns within LLMs.*
>
> ----------------------
>
> ### **Revision of Title Based on Your Feedback**
>
> After careful consideration and having carefully read your initial review, responses, and the shared agreements with `Reviewer LFdN`, **we fully agree with your comment regarding the title in your original review**. We agree that since the "Postcolonial Lens" serves primarily as a holistic theoretical framing rather than a direct variable in the experimental methodology, keeping it in the title might obscure the core empirical contribution. We have modified the title and contents of the paper accordingly; the title is now: *“Which English Do LLMs Prefer? Triangulating Structural Bias Toward American English in Foundation Models”* in the revised PDF.
>
> We cannot thank you enough for engaging with us to make the paper better. We really appreciate it.

---

### Official Review · Reviewer_nDHy · 2025-11-02

**Soundness:** 3
**Presentation:** 4
**Contribution:** 3
**Rating:** 6
**Confidence:** 4

**Summary:**

The authors of this paper investigate how large language models encode and reproduce dialectal asymmetries between American English and British English , framing these disparities through a postcolonial lens. The authors argue that this bias arises not just from downstream generations but from other aspects within the entire LLM development pipeline from training data to tokenization, etc.

**Strengths:**

- The paper is well-motivated. Most previous studies on dialectal bias primarily compare “low-resourced” dialects to Standard American English. While biases in such settings are often more severe and consequential than those in the current case studies, it is still valuable to see work examining underexplored biases toward British English.
- The framework is solid and thorough, going from corpus-level audits to tokenizer biases, and up to the “utility” of LLMs, including their generative preferences for American over British English.
- Results are not surprising, given that most stakeholders involved in developing LLMs are located in regions where Standard American English is spoken. However, they are still interesting because they provide rigorous experimental confirmation of these biases.
- A direct impact of this work is its connection to understanding how biases against British English can permeate to non-standard dialects that were adopted due to colonialism.

**Weaknesses:**

- It is not very clear in the paper how the authors control for vocabulary variants that mean the same thing across dialects but differ in form or part of speech (e.g., elevator vs. lift, apartment vs. flat). Context seems important when computing frequencies; analyzing words in isolation could overestimate or underestimate dialectal bias.
- The results show that British forms yield higher fertility for vocabulary-based differences. Given the close similarity between British and American English, tokenization may depend heavily on contextual co-occurrence patterns. Analyzing fertility in isolation might wrongly estimate the contribution of dialectal differences to tokenization biases.
- Observing small differences in tokenization parity is useful, and i appreciate the downstream experiments on generative preferences. However, from an efficiency point of view, are there really large detrimental impacts from overtokenization of British English? For example, in full sentences in full sentences with other function words, would two or three overtokenized words meaningfully increase sequence length? This might matter more for dialects with larger orthographic and lexical differences, especially those influenced by other languages in other regions, than for British vs. American English.
- I think it might be a stretch to claim that models that the large vocabulary size of Gemma guided by dialect-aware corpora will dramatically improve dialect coverage in the tokenizers. Can you expand further on this ?

**Questions:**

- Recent LLM training data are often scraped from broad web sources across the globe. If tokenizers were designed to be more British-dialect aware, would this lead to many underused tokens in the vocabulary during training?
- What are the consequences of these biases for countries that adopted British English due to colonialism? How severe might these biases be when interacting with non-standard dialects of British English? I think this is a really interesting topic that is missing from your paper, but seems relevant given your framing and paper title.

---

> ### Author Response · Authors · 2025-11-21
> **Authors' Response (Part 1/2)**
>
> Thanks a lot for your feedback. We really appreciate your effort in carefully reading the paper. Our responses to your comments are as follows.
>
> > **Comment:** It is not very clear in the paper how the authors control for vocabulary variants that mean the same thing across dialects but differ in form or part of speech (e.g., elevator vs. lift, apartment vs. flat). Context seems important when computing frequencies; analyzing words in isolation could overestimate or underestimate dialectal bias.
>
> **Response:** Context is definitely important, and this issue mainly concerns vocabulary-based differences. In our lexicon, vocabulary-based variants account for about **21%** of the pairs, and only a small subset are cases where context or part of speech is likely to materially affect interpretation, such as the examples you mention. We made this simplification because computing POS tags and context-sensitive statistics over the six pretraining corpora would be prohibitively expensive in academic setting: together, they contain more than **770 billion tokens**. At this scale, type-level frequencies for these variants tend to converge to their overall dialectal preferences, even without explicit POS control. We acknowledge this limitation and discuss it explicitly in the Limitations section.
>
> ---------------------------------------------------------
>
> > **Comment:** The results show that British forms yield higher fertility for vocabulary-based differences. Given the close similarity between British and American English, tokenization may depend heavily on contextual co-occurrence patterns. Analyzing fertility in isolation might wrongly estimate the contribution of dialectal differences to tokenization biases.
>
> **Response:** This is partly addressed in our previous response. The main objective of our triangulation framework is to identify the root cause: the presence of modeling bias. We triangulate evidence across the entire LLM development pipeline to surface structural biases in **(1)** pretraining corpora, **(2)** tokenizer representation, and **(3)** generative preferences. This setup gives us fine-grained traces of where asymmetries are introduced, amplified, and ultimately manifested in outputs, allowing RQ1, RQ2, and RQ3 to fit together into a coherent story. At the end of each RQ, we connect its findings to the preceding questions and to support concrete recommendations (`Section 8`).
>
> Our quantification of regional tokenizer behavior has **two main** goals: **(1)** to measure asymmetries (RQ2), and **(2)** to inform component-wise design recommendations (`Section 8`). For RQ2, we therefore analyze tokenizers along *two* axes. **First**, to assess representational parity at the tokenizer layer, we use fertility, which captures a mean-level view. **Second**, to obtain a more granular picture, we compute the full token-length distribution for each tokenizer, reflecting how often words are split into 1, 2, 3, or more subword units (`Lines 365–370`). We refer to this distributional diagnostic as **granularity** (`Figure 4`). It tells us how actual words are mapped into a finite vocabulary, which requires analyzing single words rather than contextual co-occurrence. This analysis directly informs how one might expand the vocabulary to make tokenizers more BrE-aware (`Lines 526–531`). We provide further elaboration on this point in the next response.
>
> --------------------------
>
> > **Comment:** I think it might be a stretch to claim that models that the large vocabulary size of Gemma guided by dialect-aware corpora will dramatically improve dialect coverage in the tokenizers. Can you expand further on this ?
>
> **Response:** Gemma achieves the lowest overall fertility, largely due to its larger vocabulary, and this is confirmed by our granularity analysis, where we quantify how often words are split into 1, 2, 3, or more subwords. This provides a clear view of how the finite vocabulary budget is allocated. For example, in Figure 4(b), Gemma maps a higher proportion of BrE variants to single tokens, and also shows strong coverage at the 2-subword level. In practice, this means many BrE forms are already present as whole words or minimally segmented units in the tokenizer. This richer coverage correlates with Gemma’s higher BrE alignment in RQ3, and we explicitly connect the RQ2 tokenizer findings to the generation results in RQ3 (`Lines 491–492`).
>
> Building on this, our granularity-based diagnostic can be used to identify BrE words that are still missing or over-fragmented and selectively inject them via controlled vocabulary expansion, as discussed in `Lines 526–531`. Given an extended vocabulary, BrE-enriched pretraining data (e.g., collected using URL-based heuristics, as outlined earlier) can then be used to continue pretraining base models to improve dialectal representation. In light of your comment, we have slightly clarified the wording of this claim in `Lines 380–383`.
>
> --------------------------------

---

> ### Author Response · Authors · 2025-11-21
> **Authors' Response (Part 2/2)**
>
> > **Comment:** What are the consequences of these biases for countries that adopted British English due to colonialism? How severe might these biases be when interacting with non-standard dialects of British English? I think this is a really interesting topic that is missing from your paper.
>
> **Response:** We agree that this is an important and previously underdeveloped aspect of our framing. Based on your recommendation (*also recommended by other reviewers*), we have added an explicit *Section 3, “Triangulating LLMs Through a Postcolonial Lens,”* in the revised manuscript, where we introduce core ideas from postcolonial theory, discuss how BrE became institutionalized in countries under British colonial rule, and explain how our triangulation framework assesses modeling bias in LLMs. We now explicitly discuss the consequences of such biases for postcolonial states in `Lines 157–159`, and we further address their implications in the RQ3 **“Key Takeaways”** box (`Lines 500–504`). Thank you for this suggestion, it helped us substantially strengthen the connection between our empirical results and the postcolonial perspective in the title.
>
> -----------------------
>
> > **Comment:** Recent LLM training data are often scraped from broad web sources across the globe. If tokenizers were designed to be more British-dialect aware, would this lead to many underused tokens in the vocabulary during training?
>
> **Response:** We discuss this explicitly in Section 8: Discussion and Recommendations, `Lines 517–524`. We propose a strategy directly related to the concern you raise. Large web-scale datasets such as Common Crawl (C4) and Dolma often include metadata such as source URLs (**Figure 10 in the Appendix**), which can be leveraged to enrich dialectal coverage. For instance, BrE coverage can be increased by selectively sampling from `.uk` domains and for World Englishes that build on BrE, such as Canadian or Indian English (e.g., `.ca`, `.in`).
>
> Synthetic data generation from larger models may also be considered; however, such generations risk defaulting to AmE. In this setting, **DiAlign** provides a safeguard by verifying whether synthetic samples align with BrE before inclusion, thereby supporting balanced and representative corpus design. These pretraining data can then be used to continue pretraining base models and improve dialectal representation in LLMs, while keeping the number of underused tokens in the vocabulary zero or minimal.
>
> -----------------------
>
> > **Comment:** However, from an efficiency point of view, are there really large detrimental impacts from overtokenization of British English? For example, in full sentences in full sentences with other function words, would two or three overtokenized words meaningfully increase sequence length?
>
> **Response:** Our goal in analyzing overtokenization is partly elaborated in the previous responses. At the level of a single sentence, we agree that the impact of two or three overtokenized words on sequence length is likely minimal. However, we adopt a broader postcolonial lens: BrE continues to hold normative prestige in many former colonies, across much of the Commonwealth, and in key European Union institutions. A deeper diagnosis of how BrE is represented in the tokenizer provides important insight into its downstream impact on generation (RQ3) and on how region-specific LLMs can be developed for local users, which is directly relevant for technical AI governance and “sovereign AI” initiatives, where national governments seek to develop LLMs tailored to their own objectives and aligned with public needs.
>
> Regarding efficiency, the impact becomes more pronounced at the document level and in long-context settings, where systematic overtokenization can visibly affect latency and per-token cost over sustained use (Petrov et al., 2023; Ahia et al., 2023), as discussed in `Lines 320–321`. This is especially important if regional companies build applications on top of current LLMs and governments deploy them in education, journalism, public administration, and law.
>
> -----------------------
>
> We truly appreciate your critical engagement, which has helped us further refine our work. If you have any remaining questions, we would be glad to address them. We hope our rebuttals and revisions adequately address your concerns, and we would be very grateful if this is reflected in your final evaluation, including any adjustment to the score you feel is appropriate.

---

### Official Review · Reviewer_LFdN · 2025-11-04

**Soundness:** 4
**Presentation:** 4
**Contribution:** 1
**Rating:** 4
**Confidence:** 3

**Summary:**

This paper examines the differences between American and British English dialects in language modelling. More specifically, the paper takes a postcolonial lens to examine this distinction (as a disclaimer, I feel unqualified to review how well this is done, as I am not entirely certain what a postcolonial lens is, exactly).

Concretely, the paper first builds a corpus of British and American lexical items which constitute minimal pairs differing in: (i) spelling (e.g., modelling vs. modeling); (ii) vocabulary use (e.g., restroom vs. lou). They then examine these words: (i) frequency in pretraining corpora; (ii) length under different tokenisers; (iii) prevalence in LLMs’ outputs. They find a consistent preference for American English across language models, but with differences across them (e.g., Gemma displays a weaker preference for American English than most other models).

**Strengths:**

This paper examines an important question, how specific dialectal differences may affect language model users. In particular, as the paper states:

> “The United Nations affirms language rights as fundamental, with Article 19 of the Universal Declaration guaranteeing the freedom to communicate in one’s language of choice.”

It is thus important that language model users be able to use these technologies using their own dialect.


This paper also builds a new corpus to perform a more specific comparison between American vs. British English.

Finally, the paper investigates statistics of these collected words in language models’: pretraining data, tokenisers, and outputs.

**Weaknesses:**

The paper’s main practical contributions are: a dataset, and three relatively “simple” comparisons between British and American English. Beyond these, the paper also contributes with its research question, i.e., with the idea to compare two major dialects of English. I think that, if published, this paper could have a reasonable impact as it documents this disparity.

While this paper addresses an important topic, however, I am unsure whether its contributions warrant an ICLR paper, so I am leaving my score as borderline reject. In particular, the paper makes no methodological contributions, and its analyses are all fairly straightforward. Further, the paper compares only two dialects of a single language. The paper is thus not comprehensive either in its experiments (i.e., only analysing simple word-level statistics) or in its objects of analysis (i.e., the set of analysed dialects). I choose this score with reservations, though, as I do believe the paper could have a reasonable impact (as said above), and I am open to increasing my score if convinced that I am missing some points of contribution.

The paper also restricts its analyses to single-word items. They justify this by stating that this “ensures consistency across analyses and is essential for the tokenizer study [RQ2 (§5)], where precise word-level comparisons are required to directly compare segmentation behavior.” However, several languages lack a clear concept of what a word is—and, as this paper investigates dialects from a postcolonial lens, using a Westernised notion of wordhood should warrant at least a short discussion. Furthermore, if two dialects exhibit systematic differences in how they segment “words”, this may lead to biased comparisons between them. Beyond that, from a technical standpoint, I see no reason why this restriction to single words is required in this paper, as all performed analyses seem to apply to multi-word expressions.

Maybe this paper's contributions could also be strengthened by expanding on what exactly makes this paper's analyses postcolonial. How does a postcolonial lens influence this study, and in which way is it better than/different from a "traditional" analysis because of this lens? Or was the postcolonial lens relevant only in the paper's choice of research question, but not in its choice of methodology?

**Questions:**

> Title: Quantifying American and British English Through a Postcolonial Lens

Could you expand on what exactly is different between “traditional” vs. “postcolonial-lens” analyses? In particular, as it applies to examining the prevalence of American vs. British Englishes in language models’ tokenisers and outputs, how does a postcolonial-lens analysis differ from a “typical” analysis. I believe several people in machine learning might similarly appreciate an exposition about this difference, so adding a dedicated discussion of this distinction to the paper could be helpful.

> Tokenizer fairness.

As the paper states, “equivalent strings can receive uneven tokenization across languages [...] (Petrov et al., 2023).” Relatedly, Lesci et al. (2025) estimate the causal effect of such uneven tokenisation on language models’ outputs, showing that a single-token word may receive up to 17 times more probability than it would if tokenised as two tokens. This effect should add to the point that these uneven tokenisations across languages will be reflected in differences in a model's quality on them. Another potentially relevant paper here is Fourotan et al. (2025), who propose a new BPE variant which aims to improve cross-lingual fairness.


> we retained only strict one-to-one word-level mappings, and excluded many-to-one

I do not understand the restriction of the analysis to single-word units. How would multi-word units affect the analyses here?


## References

* Lesci et al. (2025). Causal Estimation of Tokenisation Bias. ACL. https://aclanthology.org/2025.acl-long.1374/
* Fourotan et al. (2025). Parity-Aware Byte-Pair Encoding: Improving Cross-lingual Fairness in Tokenization. arXiv. https://arxiv.org/abs/2508.04796

---

> ### Author Response · Authors · 2025-11-21
> **Authors' Response (Part 1/3)**
>
> Thank you for your valuable feedback. We have made every effort to address your concerns and provide detailed explanations.
>
> > **Comment:** Could you expand on what exactly is different between “traditional” vs. “postcolonial-lens” analyses? In particular, as it applies to examining the prevalence of American vs. British Englishes in language models’ tokenisers and outputs, how does a postcolonial-lens analysis differ from a “typical” analysis. I believe several people in machine learning might similarly appreciate an exposition about this difference, so adding a dedicated discussion of this distinction to the paper could be helpful.
>
> **Response:** We explicitly draw on postcolonial theory, which studies how power relations created by colonialism persist after formal empire, shaping language, culture, and knowledge in both formerly colonized states and former imperial centers (Bhabha, 1994; Schneider, 2007). **The global spread of English was inseparable from British colonial expansion**: BrE was installed as the language of administration, education, and law across large parts of Africa, Asia, the Caribbean, and the Pacific, and often persisted as the official or de facto standard after independence (Acolad, 2020; Liao, 2023). It continues to hold normative prestige in many former colonies, across much of the Commonwealth, and in key European Union institutions (https://en.wikipedia.org/wiki/British_English).
>
> **At the same time, AmE dominates mass media and digital communication, and LLMs trained on web-scale internet data are likely to inherit its norms.** Our question is therefore which variety of English LLMs preferentially learn, encode, and propagate, given these different historical and institutional roles. Our postcolonial lens explicitly interprets AmE as a structurally advantaged default and BrE as a postcolonial standard embedded in institutions across many countries, and then asks how this asymmetry is reflected in pretraining data, tokenizers, and generative behavior. This framing allows us to make a controlled, high-precision comparison between two influential dialects that coexist and compete as standards, and to trace how their asymmetries arise across the entire LLM development pipeline.
>
> As governments and institutions across the globe begin to adopt these models for education, journalism, public administration, and law, understanding which standard is treated as the default becomes a critical question of linguistic equity and potential linguistic homogenization. The postcolonial lens highlights why BrE’s role in many postcolonial contexts matters and what may be lost if LLMs implicitly normalize only AmE, and how region-specific LLMs can be developed for local users, which is directly relevant for technical AI governance and *“sovereign AI”* initiatives.
>
> Based on your comment, we have dedicated a separate section in the revised paper, *“Section 3: Triangulating LLMs Through a Postcolonial Lens”* (`Lines 145–190`), where we introduce core ideas from postcolonial theory, discuss how BrE became institutionalized in countries under British colonial rule, and explain how our triangulation framework is used to assess modeling bias in LLMs. Thank you for prompting us to make this distinction more explicit.
>
> -----------------------

---

> ### Author Response · Authors · 2025-11-21
> **Authors' Response (Part 2/3)**
>
> > **Comment:** The paper also restricts its analyses to single-word items. They justify this by stating that this “ensures consistency across analyses and is essential for the tokenizer study [RQ2 (§5)], where precise word-level comparisons are required to directly compare segmentation behavior.” However, several languages lack a clear concept of what a word is—and, as this paper investigates dialects from a postcolonial lens, using a Westernised notion of wordhood should warrant at least a short discussion.
>
> **Response:** The main objective of our triangulation framework is to identify the root cause: the presence of modeling bias. We triangulate evidence across the entire LLM development pipeline to surface structural biases in **(1)** pretraining corpora, **(2)** tokenizer representation, and **(3)** generative preferences. This setup gives us fine-grained traces of where asymmetries are introduced, amplified, and ultimately manifested in outputs, allowing RQ1, RQ2, and RQ3 to fit together into a coherent story. At the end of each RQ, we connect its findings to the preceding questions and use them to support concrete recommendations (`Section 8`).
>
> The triangulation can be divided into two phases: a “pre” phase (RQ1 and RQ2) and a “post” phase (RQ3). For RQ3 (*generative preferences*), **we do not restrict our quantification to single words**. Instead, we introduce a novel scoring method, **DiAlign**, that aims to capture commonly preferred lexical, grammatical, structural, stylistic, and multi-word contrasts. We extract contiguous n-grams of length 2 to 5 to capture these multi-word expressions, as elaborated in Lines 394–400 and Section 7.1.
>
> Our quantification of regional tokenizer behavior has *two* main goals: **(1)** to measure asymmetries (RQ2), and **(2)** to inform component-wise design recommendations (`Section 8`). For RQ2, we therefore analyze tokenizers along *two* axes. **First**, to assess representational parity at the tokenizer layer, we use fertility, which captures a mean-level view. **Second**, to obtain a more granular picture, we compute the full token-length distribution for each tokenizer, reflecting how often words are split into 1, 2, 3, or more subword units (`Lines 365–370`). We refer to this distributional diagnostic as **granularity** (`Figure 4`). It tells us how actual words are mapped into a finite vocabulary, which in this analysis requires single-word units. This, in turn, directly informs how one might expand the vocabulary to make tokenizers more BrE-aware (`Lines 526–531`).
>
> In practice, the granularity-based diagnostic shows that many BrE forms are already present as whole words or minimally segmented units in the tokenizer. Building on this, it can be used to identify BrE words that are still missing or over-fragmented and selectively inject them via controlled vocabulary expansion, as discussed in `Lines 526–531`. Given an extended vocabulary, BrE-enriched pretraining data (e.g., collected using URL-based heuristics) can then be used to continue pretraining base models to improve dialectal representation.
>
> We agree that using a Westernised notion of wordhood warrants a short discussion. We now explicitly acknowledge this in the Ethics Statement (`Lines 548–552`) in the revised manuscript, noting that our analysis is English-specific and that extensions to other languages would need to adapt segmentation assumptions to local linguistic norms.
>
> ----------------------------------------------------

---

> ### Author Response · Authors · 2025-11-21
> **Authors' Response (Part 3/3)**
>
> ### **Regarding Recent Related Work**
>
> We found Lesci et al. (2025) and Fourotan et al. (2025) highly relevant to our study, and we now **cite and discuss** both in the related work section (`Lines 122–126`) in the revised manuscript. An interesting detail is that one paper’s title uses *“tokeniser”* and the other uses *“tokenizer,”* reflecting their respective choices of British vs. American spelling. Nevertheless, thank you for pointing us to these papers.
>
> ---------------------------------------
>
> ### **Paper’s Core Contributions**
>
> We believe this work is both important and impactful. The main goal of our triangulation framework is to diagnose the root cause: the presence of a previously unexplored **modeling bias** in two dominant standard varieties, and to analyze them through a postcolonial lens to demonstrate its significance. We triangulate evidence across the entire LLM development pipeline to surface structural biases in **(1)** pretraining corpora, **(2)** tokenizer representations, and **(3)** generative preferences.
>
> This holistic framework gives us fine-grained traces of where asymmetries are introduced, amplified, and ultimately manifested in outputs, allowing RQ1, RQ2, and RQ3 to fit together into a coherent narrative. At the end of each RQ, we connect its findings to the preceding questions and use them to support concrete recommendations (`Section 8`). We believe this has broader appeal because it makes the development of LLMs with regional goals for their users more actionable, and we also group the results of different LLMs by country of origin (RQ2: Table 3 and RQ3: Table 4) to give a broader view of current regional patterns.
>
> In addition to the triangulation framework and the postcolonial lens for studying LLMs, we propose a novel scoring method, **DiAlign**, to estimate dialectal preferences in LLM generations. DiAlign aims to capture commonly preferred lexical, grammatical, structural, stylistic, and multi-word contrasts (as elaborated in Section 7: RQ3). It has **broader applicability** across diverse contexts, including pretraining data audits and the filtering of both existing corpora and synthetic data (also acknowledged by **Reviewer Lvkx**). We believe our contributions will be valuable to the broader ICLR community, and your support would help bring these findings to that audience.
>
> ------------------------------
>
> Thank you once again for your valuable suggestions and insights. If you have any remaining questions, we would be glad to address them. We hope our rebuttals and revisions adequately address your concerns, and we would be very grateful if this is reflected in your final evaluation, including any adjustment to the score you feel is appropriate.

---

> ### Comment · Reviewer_LFdN · 2025-11-26
>
> I thank the authors for their responses. I think these were helpful. Correct me if I am wrong, but as I understand it now, the contribution of using a postcolonial lens here is more holistic (guiding the choice of research question and approaches used) than methodological and is, therefore, not very palpable. I have increased my score, but I have also decreased my confidence. As I see them, the paper's contributions mostly stem from its reliance on a "postcolonial lens" to perform its analyses, which I don't feel entirely equipped to evaluate.
>
> Regarding the change of title from "Quantifying American and British English Through a Postcolonial Lens" to "Triangulating the Americanization of English in LLMs Through a Postcolonial Lens". While "American English" is a technical term, there is some controversy regarding using the terms "Americanization" or "American" to mean "United States", as it appropriates the name of a continent (America) to refer to a single country. Similar to the topic of wordhood, I wouldn't raise this as an issue for most papers, but since this paper takes a "postcolonial lens", being extra careful with this kind of language may be important (e.g., see a discussion in this news article https://www.theatlantic.com/national/archive/2013/06/what-does-american-actually-mean/276999).
>
> As an unrelated note, I think Wendler et al. (2024) and a number of follow-up works are tangentially related to this analysis, and could be interesting to discuss.
>
> Wendler et al. (2024).  Do Llamas Work in English? On the Latent Language of Multilingual Transformers. https://arxiv.org/abs/2402.10588

---

> > ### Author Response · Authors · 2025-11-28
> > **Authors' Response to Reviewer Response (Part 2/2)**
> >
> > > I think Wendler et al. (2024) and a number of follow-up works are tangentially related to this analysis, and could be interesting to discuss.
> >
> > **Response:** We are sincerely grateful to you for directing us to Wendler et al. (2024). Upon review, we agree that this work is highly relevant and provides a crucial **mechanistic perspective** on our findings.
> >
> > We found that it aligns closely with Zhao et al. (2024), who similarly observe that multilingual models process inputs by mapping them to a *“latent pivot”* or English subspace before decoding. We have explicitly linked them to our findings in the Results & Analysis of RQ3 (`Section 7`) in the revised manuscript (`Lines 498–500`).
> >
> > While Wendler and Zhao establish that the latent pivot is English, our triangulation findings (RQ1–RQ3) suggest this pivot is mostly American English. **This offers a good explanation** for why the models in our study struggle to generate British English even when explicitly prompted: they are essentially fighting against the gravitational pull of a structural "American English default" within their latent reasoning space.
> >
> > Thank you for this excellent suggestion; it has significantly strengthened the interpretation of our generative outputs (RQ3).
> >
> > 1. Wendler et al., 2024, Do Llamas Work in English? On the Latent Language of Multilingual Transformers, ACL (https://aclanthology.org/2024.acl-long.820/).
> > 2. Zhao et al., 2024, How do Large Language Models Handle Multilingualism?, NeurIPS 2024 (https://openreview.net/forum?id=ctXYOoAgRy).
> >
> > -------------------------
> >
> > Thank you again for your time, your insights, and for helping us to improve the paper. We truly value the collaborative spirit of this exchange; it has allowed us to sharpen our arguments and better contextualize our findings within the broader literature. We are very pleased with how the manuscript has evolved based on your feedback. If you have any further questions or suggestions, we would be happy to address them.

---

> ### Author Response · Authors · 2025-11-28
> **Authors' Response to Reviewer Response (Part 1/2)**
>
> We sincerely thank you for your detailed feedback and for **``raising your score to 6 based on our initial rebuttals``**. We deeply appreciate your active engagement, particularly your thoughtful suggestions regarding the title and the relevant literature. This exchange has been very productive, and we are grateful for the opportunity to refine our work through our conversation.
>
> ----------------------
>
> ### **On Holistic and Theoretical Postcolonial Lens**
>
> You are entirely correct: its contribution is indeed holistic and theoretical rather than strictly methodological. We have refined the manuscript to reflect this exact understanding. Below, we clarify our approach and detail how your comments have helped us to refine our work:
>
> Our paper has a central question, **“Which English Do LLMs Prefer?”**, and uses a **triangulation methodology** (Data Audit + Tokenizer Analysis + Generation Evaluation) to arrive at our main finding: “Structural Bias Toward American English.” The “Postcolonial Lens” is a theoretical and holistic view that helped us narrow the scope and enable a controlled, high-precision comparison where we position AmE as the structurally advantaged default and BrE as a widely institutionalized yet comparatively marginalized variety. Moreover, this framing interprets the study’s broader significance: BrE retains normative prestige in many former colonies, where it remains embedded in governance, education, and law. It is also the standard of EU institutions and underpins “Commonwealth English,” actively promoted by the UK across **more than 100 countries** through initiatives such as the Oxford Dictionary, the British Council, and IELTS. We also cited sociolinguistic research and other papers in the Introduction and Section 3 to motivate and back this up. **For a quick reference, you may visit this [Wikipedia link](https://en.wikipedia.org/wiki/British_English#:~:text=Globally%2C%20countries%20that%20are%20former,world%20and%20operates%20in%20over%20100%20countries.).**
>
> On the other hand, AmE is dominant in digital communication through mass media and technological platforms (Gonçalves et al., 2018). **We acknowledge that this prevalence is primarily driven by population size and the sheer volume of US-generated content.** However, sociolinguistic research (Milroy & Milroy, 1999; Lippi-Green, 2012) shows that **power dynamics** are involved in transforming this statistical majority into a *de facto* global norm. Nevertheless, while the training of LLMs relies solely on massive digital corpora largely drawn from the internet, **these models are used by users across the globe**. The “Postcolonial Lens” highlights the broader significance of this study (also acknowledged by `Reviewer nDHy`), particularly in the context of education, journalism, government, and legal texts, where there is a risk of linguistic homogenization through LLMs.
>
> To this end, we triangulate evidence across the entire LLM development pipeline to surface structural biases in **(1)** pretraining corpora, **(2)** tokenizer representations, and **(3)** generative preferences. This setup gives us fine-grained traces of where asymmetries are introduced, amplified, and ultimately manifested in outputs, allowing RQ1, RQ2, and RQ3 to fit together into a coherent story. At the end of each RQ, we connect its findings to the preceding questions and use them to support concrete recommendations (Section 8), which makes the development of LLMs with regional goals for their users more actionable. *We also group the results of different LLMs by country of origin (e.g., USA, UK, France, Italy, China, UAE) (RQ2: Table 3 and RQ3: Table 4) to give a broader view of current regional patterns within LLMs.*
>
> ### **Revision of Title Based on Your Feedback**
>
> After careful consideration of your insights regarding the term "Americanization," combined with your observation that the "Postcolonial Lens" acts as a holistic framing rather than a direct experimental variable, a point also shared by `Reviewer Lvkx`, we have modified the title in the revised PDF.
>
> We agree that since the lens serves primarily as a theoretical framework as described earlier, keeping it in the title might obscure the core empirical contribution. We have updated the manuscript accordingly, and the title is now: *“Which English Do LLMs Prefer? Triangulating Structural Bias Toward American English in Foundation Models”.*

---

### Author Response · Authors · 2025-11-21
**General Response to All Reviewers**

We thank all the reviewers for their thoughtful comments and constructive suggestions. We are encouraged that several reviewers recognized the importance and potential impact of our work. Based on the reviewer comments, we have slightly revised the title to better reflect our core contributions (“Triangulating”) and our central finding (“Americanization”) to:
*“Which English do LLMs Prefer? Triangulating the Americanization of English in LLMs Through a Postcolonial Lens.”*


Based on the reviewers’ suggestions, we have also dedicated a separate section in the revised paper, **“Section 3: Triangulating LLMs Through a Postcolonial Lens”** (`Lines 145–190`), where we introduce core ideas from postcolonial theory, discuss how BrE became institutionalized in countries under British colonial rule, and explain how our triangulation framework is used to assess modeling bias in LLMs.


**We have uploaded the revised manuscript** with changes highlighted in `dark green`, given the **10-page main-content limit** for the rebuttal. We have taken great care to address the questions and comments raised and respond to individual reviewer comments. If you have any further questions or suggestions, we would be happy to address them.


Once again, thank you for your valuable feedback, which has helped us strengthen the paper.


Best regards,

The Authors

---

### Author Response · Authors · 2025-11-30
**Summary of Discussion, Consensus, and Revisions for AC**

Dear AC and Reviewers,

We sincerely thank the reviewers for their thoughtful comments and constructive suggestions. We are pleased that the reviewers have acknowledged the **importance and potential impact of our work**.

We have taken great care to address all concerns raised by the reviewers. During the discussion phase, we engaged in a highly productive dialogue with Reviewers `Lvkx` and `LFdN`. Based on our rebuttals, `Reviewer LFdN` **raised the score from 4 to 6**. We also addressed the remaining minor concerns raised by both reviewers during conversation. *We believe that had the discussion period continued, our responses would have further clarified our contributions for all reviewers.*

This engagement has led to a **significantly improved manuscript**. The key changes include:

1. **Title Revision:** Based on the consensus feedback from Reviewers `Lvkx` and `LFdN`, we have updated the title to: *“Which English Do LLMs Prefer? Triangulating Structural Bias Toward American English in Foundation Models”.*
2. **Manuscript Updates:** We have incorporated **all reviewer feedback** into the revised manuscript, important changes are highlighted in **dark green** in the uploaded PDF.

In light of these improvements and the positive trajectory of the discussion, **we respectfully request that the AC** consider the full context of our responses alongside the reviews and the potential of our work to encourage valuable community discussion during the meta-review stage.

Thank you for your time and consideration.


Best regards,

The Authors

---

### Meta-Review · Area_Chair_CjmU · 2026-01-05

**Summary:**

Despite overall positive scores, reviewers outlined several major concerns that many of them shared:
- Postcolonial lens: Most reviewers noted that the methodology has little to do with postcolonial theory.
- Limited novel contributions/analyses: Most reviewers mentioned some aspect of this concern (e.g. considering only two dialects of a single language, little methodological/insight novelty over other works that have looked at individual components like language biases in tokenization)
- Topical fit: Reviewers noted that this submission may be a stronger topical fit for audiences at venues like ACL or COLM.

**Reviewer Concerns:**

This is very much a difficult, borderline decision, as the reviewers appreciated many aspects of the submission despite having major concerns outlined in their reviews. The authors did make an effort to address many of the concerns, but I believe the three major concerns outlined above remain at least partially unresolved (postcolonial lens seems to be mostly motivational rather than methodological; contribution novelty is limited over existing work examining language biases in LLM training data, tokenizers, and generation).

**Reviewer Scores:**

Authors mention that Reviewer LFdN had changed their scored from a 4 to a 6 before the freeze, which I find plausible. The remaining reviewers are not likely to have changed their scores.

---

### Decision · Program_Chairs · 2026-01-26

Reject